# TVSPrune - Pruning Non-discriminative filters via Total Variation separability of intermediate representations without fine tuning

**Chaitanya Murti**[1]**, Tanay Narshana**[2*]**, and Chiranjib Bhattacharyya**[1,3]

Robert Bosch Centre for Cyber-Physical Systems, Indian Institute of Science[1]
Observe.AI[2]
Department of Computer Science and Automation, Indian Institute of Science.[3]
`{mchaitanya, chiru}@iisc.ac.in, tanay.narshana@gmail.com`

## Abstract

Achieving structured, data-free sparsity of deep neural networks (DNNs) remains an open area of research. In this work, we address the challenge of pruning filters without access to the original training set or loss function. We propose the *discriminative filters hypothesis*, that well-trained models possess discriminative filters, and any non discriminative filters can be pruned without impacting the predictive performance of the classifier. Based on this hypothesis, we propose a new paradigm for pruning neural networks: distributional pruning, wherein we only require access to the distributions that generated the original datasets. Our approach to solving the problem of formalising and quantifying the discriminating ability of filters is through the total variation (TV) distance between the class-conditional distributions of the filter outputs. We present empirical results that, using this definition of discriminability, support our hypothesis on a variety of datasets and architectures. Next, we define the LDIFF score, a heuristic to quantify the extent to which a layer possesses a mixture of discriminative and non-discriminative filters. We empirically demonstrate that the LDIFF score is indicative of the performance of random pruning for a given layer, and thereby indicates the extent to which a layer may be pruned. Our main contribution is a novel one-shot pruning algorithm, called TVSPrune, that identifies non-discriminative filters for pruning. We extend this algorithm to IterTVSPrune, wherein we iteratively apply TVSPrune, thereby enabling us to achieve greater sparsity. Last, we demonstrate the efficacy of the TVSPrune on a variety of datasets, and show that in some cases, we can prune up to 60% of parameters with only a 2% loss of accuracy without any fine-tuning of the model, beating the nearest baseline by almost 10%. Our code is available **here**[1].

## 1 Introduction

Deep neural networks are, in general, highly overparameterized, leading to significant implementation challenges in terms of reducing inference time, power consumption, and memory footprint. This is especially crucial for deployment on real world, resource constrained devices (Molchanov et al., 2019b; Prakash et al., 2019). A variety of solutions have been proposed to solve this problem, which can broadly be grouped into quantization, sparsification or pruning, knowledge distillation, and neural architecture search (NAS) (Hoefler et al., 2021).

Pruning can be further divided into unstructured pruning, wherein individual parameters are set to zero, or structured pruning, wherein entire filters or channels are removed from the architecture (Hoefler et al., 2021). Structured pruning yields immediate improvements in inference time, memory footprint, and power consumption, without requiring any specialized software frameworks. Unstructured pruning, on the other hand, typically yields models that are significantly more sparse than those

---

[*]Work done at the Department of Computer Science and Automation, Indian Institute of Science.
[1]Link to github: `https://github.com/chaimurti/TVSPrune`

obtained with structured pruning, but do not provide the same improvements in inference time without specialized sparse linear algebra implementations (Hoefler et al., 2021; Blalock et al., 2020).

In this work, we consider the problem of pruning CNNs without access to the training set or loss function. Data-free pruning is an important problem due to concerns such as privacy and security Yin et al. (2020), as well as the cost of retraining models (Tanaka et al., 2020; Hoefler et al., 2021). We offer a new perspective to this problem, which we call *distributional pruning*, wherein *we have no access to the training data or the loss function*, but have access to data distribution, either through it's moments, or through additional samples separate from the training set.

To facilitate distributional pruning, two crucial questions need to be answered. First, *what makes a filter valuable to the classification performance of the model?*, and second, *how do we characterize which layers can be effectively sparsified?* To answer these questions, we first identify *discriminative filters*, which are filters with class-conditional outputs that are well-separated in terms of the total variation. We propose the *discriminative filters hypothesis*, which states that well-trained models possess a mix of discriminative and non discriminative filters, and that discriminative filters are useful for generalization of the classifier. Based on this hypothesis, discriminative filters are useful for classification purposes whereas non-discriminative filters are not, thus allowing us to prune the latter. Furthermore, layers that possess a mix of discriminative and non-discriminative filters can be effectively pruned, thereby providing a method to identify difficult-to-prune layers. We formally state our contributions below.

1. We begin by proposing a quantitative measure of discriminative ability of filters in terms of the TV distance between their class-conditional outputs. Specifically, we say a filter is TV-separable if the pairwise minimum TV distance between class-conditional distributions of the outputs is larger than a given threshold. If the class conditional distributions are gaussian - a common assumption as noted in Wong et al. (2021); Wen et al. (2016) - we can compute the Hellinger distance based lower bound to estimate whether filters are TV-separable using easily computed class-conditional moments. We describe this in section 4.

2. We produce the empirical observation that the classwise outputs of at least some filters present in CNNs that generalize well are TV-separable, and that untrained models, or models that generalize poorly do not possess discriminative filters; these are presented in section 7. Based on these observations, in section 2, we propose the *discriminative filters hypothesis*, which states that *well-trained convolutional neural networks possess a mix of discriminative and non-discriminative filters, and discriminative filters are useful for classification whereas the latter are not*. We use this hypothesis to motivate a distributional approach to pruning.

3. Based on the discriminative filters hypothesis, we aim to use TV separation to identify which filters to prune in a model. We assume the class-conditional distributions are Gaussian, and, using the Hellinger lower bounds discussed in section 4, we compute lower bounds on the TV-separation for each filter. We identify important filters (those that cannot be pruned) as those filters with the Hellinger lower bound of the TV-separation that are greater than a *separation threshold*; those filters that are not discriminative with respect to the separation threshold can be pruned.

4. As noted in Hoefler et al. (2021); Liebenwein et al. (2019), some layers are more difficult to prune than others. We address the problem of identifying which layers can be effectively pruned using TV-separability. Based on the discriminative filters hypothesis, a layer can be effectively pruned if it possesses a mixture of discriminative and non-discriminative filters. Thus, in section 5, we propose an informative heuristic, which we call the LDIFFscore, that quantifies the extent to which a layer possesses a mixture of discriminative and non-discriminative filters. We empirically validate this heuristic in section 7.

5. We use TV-separability and LDIFF scores to develop TVSPRUNE, a layer-wise, threshold based method for structured pruning, requiring no fine tuning and only the class-conditional moments of the outputs of each filter. We also extend this algorithm to an iterative variant, ITERTVSPRUNE, which enables superior sparsification of the model. We formally state these algorithms in section 6, in Algorithms 1 and 2. We show that on the CIFAR-10 dataset, our method achieves over 40% sparsification with minimal reduction in accuracy on VGG models without any fine tuning; furthermore, our method outperforms contemporary methods such as (Sui et al., 2021; Molchanov et al., 2019b) in this regime.

Our paper is organized as follows. In Section 2, we outline our problem statement. In Section 3, we detail the notation used in the paper, and review Total Variation distances and the associated Hellinger distance bounds, and in Section 4, we describe how we utilize the TV-distance to measure the discriminative capacity of filters. In Section 5, we propose the LDIFF score for quantifying the difficulty in pruning a given layer, and in Section 6, we present the TVSPRUNE and ITERTVSPRUNEalgorithms. In Section 7, we detail our experimental results. We produce a detailed related work section in Appendix A. We also provide additional discussions in B, variants of the LDIFF score and the TVSPRUNE algorithms in sections C and D, and a variety of additional empirical results in section E.

## 2   Filter Pruning without Training Data and Fine Tuning

Structured pruning can be thought of as two specific optimization problems: finding the most accurate model that satisfies a sparsity model, and finding the sparsest network that satisfies an accuracy constraint. We define the latter problem below.

$$\mathcal{W}^* = \arg\min_{\mathcal{W}} \left\{ \|\mathcal{W}\|_0 \mid f(\mathcal{W}) \le t, \ t > 0 \right\}. \tag{P}$$

Our goal is to solve equation P using structured pruning techniques; that is, we aim to remove entire filters at each pruning iteration. As mentioned previously, typical methods to solve this problem require extensive retraining to overcome the losses in accuracy. In this work, we consider the scenario wherein neither the training data nor the loss function are available during the pruning process.

Motivated by empirical results, we see that models that generalize well tend to have filters with well-separated output features (in the distributional sense), whereas models that do not generalize well possess filters with output features that are not as well separated. We refer the reader to Section 7.1 for these results. This prompts us to formulate the following hypothesis.

**Hypothesis 1 (Discriminative Filters Hypothesis).** *"Models that generalize well possess a rich mixture of filters that discriminate between classes and those that don't. Furthermore, we hypothesize that for generalization purposes, it is sufficient to retain the filters that discriminate between classes well, in those layers that have a large variation in discriminative ability."*

Note that we also observe that some layers have filters with output features that cannot discriminate well, and yet cannot be pruned.

## 3   Preliminaries and Background

### 3.1   Notation

Let $[n] = \{1, \cdots, n\} \subset \mathbb{N}$. We the define parameters of a neural network with $L$ as $\mathcal{W} = \{W_1, \cdots, W_L\}$, where each $W_i = \{W_{i,j}\}_{j=1}^{N_l}$ is a collection of tensors. For a convolutional neural network (CNN), each $W_l \in \mathbb{R}^{N_{in}^l \times N_{out}^l \times K^l \times K^l}$ . In this formulation, the $i$th filter of layer $l$ is a tensor $f_i^l \in \mathbb{R}^{N_{in} \times K \times K}$, and the $j$th kernel in this filter is a matrix $f_{i,j}^l \in \mathbb{R}^{K \times K}$. Note that given a model, we can use the pair $(l, j)$ to identify a filter - $l$ denotes the layer, and $j$ the filter index.

We vectorize this model as follows. First, suppose the vectorized input to layer $l + 1 \in [L]$ is $\phi^l(X) \in \mathbb{R}^{m^2 N_{in}}$, where $N_{in}$ is the number of channels and $m$ is the dimension of the input (say, the number of pixels in each dimension of an image). Note that we ignore the superscript indicating the layer for the sake of brevity. Suppose each filter in layer $l + 1$ has kernel size $K \times K$, that is, for each $j \in [J]$, the we have the vectorized $W_j^l \in \mathbb{R}^{K^2 N_{in}}$. Next, we assume that there are $P$ patches, with $\phi_p^l(X)$ being the $p$th patch in the input. In the sequel, we drop the layer superscript for convenience. In this work, we assume we draw samples $(X, u)$, where $X$ is the datum and $u$ is the class label, from a distribution $\mathcal{D}$. We write $\mathcal{D}^\alpha$ is the class conditional distribution of class $\alpha$. Here $X \in \mathbb{R}^N$ for some integer $N$ is the input, and $u \in [N_{classes}] \subset \mathbb{N}$ is the integer class.

## 3.2 REVIEW OF TOTAL VARIATION DISTANCE

In this section, we formally define the Total Variation and Hellinger distances. We define this formally in the sequel. The results that follow are discussed in, say, Kraft (1955).

**Definition 1.** *Let $\mathbb{Q}_1$ and $\mathbb{Q}_1$ be two probability measures supported on $\mathbb{R}^d$. We define the Total Variation Distance TV as*

$$\text{TV}(\mathbb{Q}_1, \mathbb{Q}_2) = \sup_{A \subset \mathbb{R}^d} |\mathbb{Q}_1(A) - \mathbb{Q}_2(A)|$$

Unfortunately, no closed form expression exists for the Total Variation distance, even when $\mathbb{Q}_1, \mathbb{Q}_2$ are Gaussian. This motivates us to use the Hellinger distance, which we define below.

**Definition 2.** *Let $\mathbb{Q}_1, \mathbb{Q}_2$ be two probability measures supported on $\mathbb{R}^d$, and let $q_1$ and $q_2$ be the corresponding densities. We define the squared Hellinger distance as*

$$\text{HELLD}^2(\mathbb{Q}_1, \mathbb{Q}_2) = \frac{1}{2} \int_{\mathbb{R}^d} \left( \sqrt{q_1(x)} - \sqrt{q_2(x)} \right)^2 dx$$

Crucially, the Hellinger distance provides us upper and lower bounds on the TV-Distance.

$$\text{HELLD}^2(\mathbb{Q}_1, \mathbb{Q}_2) \leq \text{TV}(\mathbb{Q}_1, \mathbb{Q}_2) \leq \sqrt{2} \, \text{HELLD}(\mathbb{Q}_1, \mathbb{Q}_2). \tag{1}$$

If we have $\mathbb{Q}_1 \sim \mathcal{N}(\mu_1, \sigma_1^2 I)$ and $\mathbb{Q}_2 \sim \mathcal{N}(\mu_2, \sigma_2^2 I)$ are Gaussian, we get

$$\text{HELLD}^2(\mathbb{Q}_1, \mathbb{Q}_2) = 1 - \left( \frac{2\sigma_1 \sigma_2}{\sigma_1^2 + \sigma_2^2} \right)^{\frac{d}{2}} e^{-\frac{\Delta}{4}}, \quad \Delta = \frac{\|\mu_1 - \mu_2\|^2}{\sigma_1^2 + \sigma_2^2} \tag{2}$$

## 4 MEASURING DISCRIMINATIVE ABILITY OF FILTERS

In this work, we take a distributional view of the discriminative ability of filters. Previously, we have described the TV-distance, and the Hellinger lower bound that can be efficiently estimated. In this section, we apply the TV-distance to arriving at a means to quantify the discriminative ability of individual filters by using the class-conditional distributions of the outputs. We begin by recasting the dot products used in the convolution operations with vectorized filters and feature maps.

Define $y_{j,p}(X)$ to be the dot product of the $j$th filter and the $p$th patch of $\phi^{l-1}(X)$; that is

$$y_{l,j,p}(X) = \langle \phi_p^{l-1}(X), W_j^l \rangle.$$

Following from this, we can write

$$Y_{l,j}(X) = \langle \Phi^{l-1}(X), W_j^l \rangle, \tag{3}$$

where $\Phi^{l-1}(X) = [\phi_1^{l-1}(X), \cdots, \phi_P^{l-1}(X)]$ We define the Class Conditional Means and Variances for $Y(X)$, the output of a given filter as follows.

$$\bar{Y}^\alpha = \mathbb{E}_{(X,u) \sim \mathcal{D}^\alpha}[Y(X)] \text{ and } \sigma_{l,j,\alpha}^2 = \mathbb{E}_{(X,u) \sim \mathcal{D}^\alpha}\left[ \|Y(X) - \bar{Y}^\alpha\|^2 \right] \tag{4}$$

**Assumption 1.** *We assume that the class conditional distributions of each filter is Gaussian, that is $Y_{l,j}(X|\alpha) \sim \mathcal{N}(\bar{Y}_{l,j}^\alpha, \sigma_{l,j,\alpha}^2)$*

This assumption enables us to use the closed form expressions described in equation 2. While this assumption is a strong one, it is well motivated in the literature. The assumption that class conditional distributions are gaussian is used in a variety of settings, including Sun et al. (2020); Lee et al. (2020); Seetharaman et al. (2019). Our goal is to quantify the discriminative ability of the outputs of the individual filters, and to do so, we use the TV-Distance between the class conditional distributions. Assumption 1 enables us to estimate lower bounds on the TV distance between any two class-conditional distributions of a filter's output easily, using only samples from that distribution. With that in mind, we define the Minumum TV Separation **(MinTVS)** between classes of a given filter.

**Definition 3.** *Suppose Asssumption 1 holds. Let $\mathcal{D}_{l,j}^\alpha$ be the class-conditional distribution of $Y_{l,j}(X)$ where $(X, u) \sim \mathcal{D}^\alpha$. For each layer $l$ and filter $j$, and each pair $\alpha, \beta$, we define the Minimum TV Separation of filter $j$ in layer $l$ as*

$$\text{MinTVS}(l, j) = \min_{\alpha, \beta} \text{ TV}\left(\mathcal{D}_{l,j}^\alpha, \mathcal{D}_{l,j}^\beta\right) \geq \min_{\alpha, \beta} \text{HELLD}^2\left(\mathcal{D}_{l,j}^\alpha, \mathcal{D}_{l,j}^\beta\right). \tag{5}$$

Furthermore, since the harmonic mean in equation 2 is upper bounded by 1, we have

$$\text{MinTVS}(l, j) \geq 1 - \exp\left(-\min_{\alpha, \beta} \Delta_{l,j}^{\alpha,\beta}/4\right), \quad \Delta_{l,j}^{\alpha,\beta} = \|\bar{Y}_{l,j}^\alpha - \bar{Y}_{l,j}^\beta\|^2 \left(\sigma_{l,j,\alpha}^2 + \sigma_{l,j,\beta}^2\right)^{-1}$$

thus allowing us to use $\Delta_{l,j}^{\alpha,\beta}$ as a surrogate for $\text{MinTVS}(l, j)$ during experimentation. Using this quantity, we define TV-Separability with respect to some $\eta > 0$, which we call the *separation threshold*.

**Definition 4.** *For each layer $l$ and filter $j$, and separation threshold $\eta > 0$ we say the filter is $\eta$-TV Separable, or $\text{TVSEP}(\eta)$ if*

$$\text{MinTVS}(l, j) \geq \eta.$$

Since we cannot directly compute the pairwise minimum of the TV-separations between the class-conditional distributions of a filter's output, we instead rely on the lower bound proposed above. The value of $\eta$ that indicates a "well separated" filter will vary from dataset to dataset, or architecture to architecture; for instance, for the VGG models trained on CIFAR10 used in our experiments, we find $\eta = 0.05$ is useful.

## 5 IDENTIFYING DIFFICULT-TO-PRUNE LAYERS AND THE LDIFF SCORE

As noted in Hoefler et al. (2021); Liebenwein et al. (2019), in a given model, some layers can be effectively pruned, and others cannot. In this section, we investigate the problem of identifying which layers of a given neural network can be pruned effectively, through the lens of the discriminative filters hypothesis. Specifically, we argue that discriminative filters contribute to generalization performance whereas non-discriminative filters do not, and argue that we can effectively prune layers containing a mix of discriminative and non discriminative filters. We desire an informative heuristic that can (a) indicate the extent a layer can be randomly pruned, thus informing pruning ratios for random pruning; and (b) indicate whether or not a layer should be pruned at all. Note that we leverage the latter purpose in our proposed algorithms in subsequent sections.

We define LDIFF, a heuristic score assigned to each layer based on the TV separations of the constituent filters to the aggregate TV-separation and a given separation threshold $\eta$. Our goal is to capture the mix of discriminative and non-discriminative filters in a single heuristic indicator. Thus, given a separation threshold $\eta$, suppose the fraction of discriminative filters is $\tau(\eta)$. We need a function that penalizes $\tau(\eta)$ close to either 1 or 0 (as having no discriminative filters in a layer indicates difficulty in pruning). With these requirements in mind, we define

$$\text{LDIFF}(l, \eta) = 4\tau(\eta)(1 - \tau(\eta)) \tag{6}$$

If this quantity is close to 1, then that layer has a mix of discriminative and non-discriminative filters. If not , then there is either a majority of discriminative filters, or non-discriminative filters; in either case, pruning that layer would be challenging. Thus, this heuristic can also be used to inform random pruning ratios. For instance, we may choose $\gamma \text{LDIFF}(l, \eta)$, for suitable $\gamma \in (0, 1)$, to be the fraction of filters to be randomly pruned. Further refinements of this heuristic include layerwise thresholds, and using statistics of the $\text{MinTVS}$ scores to determine discriminability. We refer readers to Section C for further discussion on such refinements, as well as empirical support for the use of LDIFF scores for indicating pruneability of layers.

## 6 DISTRIBUTIONAL APPROACHES TO STRUCTURED PRUNING

In this section, we discuss our solution to the problem of structured pruning of neural networks without access to the loss function or the original data, and only with access to the distributions of the

outputs of the individual filters and layers. Our proposed algorithm has three steps. First, for a given separation threshold $\eta$, we estimate the $\eta$-TV-separability of the filter outputs; second, we decide which layers we can prune; third, in those layers which we can prune from, we remove all filters with TV-separability less than $\eta$.

| **Algorithm 1:** TVSPRUNE | **Algorithm 2:** ITERTVSPRUNE |
|---|---|
| **Input:** Dataset $\hat{\mathcal{D}} = \{X_i, u_i\}_{i=1}^M$, Pretrained CNN with parameters $\mathcal{W} = (W_1, \cdots, W_L)$,TVSEP threshold $\eta$, layer difficulty threshold $\nu$ | **Input:** Dataset $\hat{\mathcal{D}} = \{X_i, u_i\}_{i=1}^M$, Pretrained CNN $\mathcal{W} = (W_1, \cdots, W_L)$, initial TVSEP threshold $\eta$, LDIFF threshold $\nu$, $\delta_\eta > 0$, accuracy threshold $t$, $\eta_{\min} < \eta$. |

**Algorithm 1:** TVSPRUNE

**Input:** Dataset $\hat{\mathcal{D}} = \{X_i, u_i\}_{i=1}^M$, Pretrained
    CNN with parameters
    $\mathcal{W} = (W_1, \cdots, W_L)$,TVSEP threshold $\eta$,
    layer difficulty threshold $\nu$
Compute $\bar{Y}_j^{l,\alpha}$, $\sigma_{l,j,\alpha}^2$ for all $l$, $j$, and $\alpha$ .
Compute $\Delta_{l,j}^{\alpha,\beta}$ for each $l, j, \alpha, \beta$.
**for** $l \in [L]$ **do**
    Compute MinTVS$(l,j)$ for all $l, j$
    Compute LDIFF$(l)$
    **if** LDIFF$(l) < \nu \vee$ LDIFF$(l) = 0$ **then**
        **for** $j \in [N_{out}^l]$ **do**
            **if** MinTVS$(l,j) < \eta$ **then**
                $W_j^l \leftarrow \mathbf{0}$

**Output:** $\hat{\mathcal{W}} = (\hat{W}_1, \cdots, \hat{W}_L)$, where
    $\mathrm{supp}(\hat{W}_l) \leq \mathrm{supp}(W_l) \; \forall \, l$
**return** $\hat{\mathcal{W}}$

**Algorithm 2:** ITERTVSPRUNE

**Input:** Dataset $\hat{\mathcal{D}} = \{X_i, u_i\}_{i=1}^M$, Pretrained
    CNN $\mathcal{W} = (W_1, \cdots, W_L)$, initial
    TVSEP threshold $\eta$, LDIFF threshold $\nu$,
    $\delta_\eta > 0$, accuracy threshold $t$, $\eta_{\min} < \eta$.
Set $k = 0$, $\mathcal{W}_{(0)} = \mathcal{W}$
**while** $\eta \geq \eta_{\min}$
    $\tilde{W} = $ TVSPRUNE$(\mathcal{D}, \mathcal{W}_{(k)}, \eta, \nu)$
    **if** $f(\tilde{\mathcal{W}}) \leq t$ **then**
        $\mathcal{W}_{(k+1)} \leftarrow \tilde{\mathcal{W}}$
    **else**
        $\eta \leftarrow \eta - \delta_\eta$
        $\mathcal{W}_{(k+1)} \leftarrow \mathcal{W}_{(k)}$
    $k \leftarrow k + 1$

**Output:** $\hat{\mathcal{W}} = (\hat{W}_1, \cdots, \hat{W}_L)$, $f(\hat{\mathcal{W}}) \leq t$,
    $\mathrm{supp}(\hat{W}_l) \leq \mathrm{supp}(W_l) \; \forall \, l$
**return** $\hat{\mathcal{W}}$

## 6.1 THE TVSPRUNE ALGORITHM - VARIABLE PRUNING RATIOS PER LAYER

The goal of this algorithm is to leverage the discriminative ability of filters to decide whether a given filter should be pruned. Thus far, we have defined the MinTVS value, which is a lower bound on the least classwise TV-separation of a filter's output, as well as the LDIFF score, which, in a sense, quantifies the extent to which a layer can be sparsified.

The algorithm begins by computing the TV-separability of all the filters using $\{\mathrm{MinTVS}(l,j)\}_{j=1}^{N_{out}^l}$ as defined in equation 10. Next, we compute the LDIFF values as described in equation 6. We then check the LDIFF score agains the threshold $\nu$, and thereby decide whether or not to prune the given layer. Then, all filters in each prunable layer (that satisfies LDIFF$(l) > \nu$) that are not $\eta$-TV Separable be pruned. The value of $\eta$ dictates the aggressiveness of the pruning as it is a measure of how "well separated" we desire - if $\eta$ is small, fewer filters are pruned. This approach is useful since it ensures that the pruning ratio varies from layer to layer, and that we prune those layers that are relatively difficult to sparsify far less aggressively.

Furthermore, we observe that some layers are dramatically more difficult to prune from than others. Therefore, we use the LDIFF scores to decide which layers to prune. We now present the TVSPRUNE algorithm. We assume that we have a labeled dataset $\hat{\mathcal{D}}$ of pairs $(X, u)$ that was not used for training. The TVSPRUNE algorithm then uses this dataset to compute the moments $\bar{Y}_{l,j}^\alpha$, $\sigma_{l,j,\alpha}^2$ for each layer $l$, filter $j$, and class $\alpha$. Using this data, TVSPRUNE then computes MinTVS$(l,j)$ for each $l,j$; after this, the mean $m_l$ and standard deviation $\sigma_l$ are computed for the TVSEPvalues $\{\mathrm{MinTVS}(l,j)\}$ for each layer $l$. After these are computed, those filters that satisfy MinTVS$(l,j) \leq \eta$ are pruned.

## 6.2 ITERATIVE PRUNING WITH TV SEPARATION

We motivate this section by noting that the TVSPRUNE algorithm does not give us control over the accuracy of the pruned model, or the extent that a model is sparsified. Therefore, it is unsuitable to solve either equation P . In this section, we describe ITERTVSPRUNE, an iterative algorithm that builds upon TVSPRUNE, and which attempts to solve equation P. We also propose a variant of ITERTVSPRUNE that attempts to solve the problem of finding the most accurate model given a sparsity budget, without training data or access to the loss function. We refer readers to the supplemental material for this algorithm description.

The ITERTVSPRUNE algorithm builds upon the fact that the distribution over $Y^l(X)$ changes each time the previous layers are pruned. Thus, we may recompute the TV-Separation values, and iteratively prune the model. Note that as was the case with TVSPRUNE, this algorithm does not require any fine-tuning, and does not even require the training data.

The ITERTVSPRUNE algorithm takes an initial TVSEP threshold $\eta$, a convolutional neural network with parameters $\mathcal{W}$, an LDIFF threhsold $\nu$, and an accuracy threshold $t$. At each step $k$, we run the TVSPRUNE algorithm given the current parameters $\mathcal{W}_{(k)}$ and a current TVSEP threshold $\eta_k$; that is $\bar{\mathcal{W}} = \text{TVSPRUNE}(\mathcal{W}_{(k)}, \eta_k, \mathcal{D})$. After running TVSPRUNE, we check the accuracy of the pruned model - if $f(\bar{\mathcal{W}}) \leq t$, then we repeat the pruning step, else we reduce $\eta$ and run TVSPRUNE with $\mathcal{W}_{(k)}$ and the new value of $\eta$. The utility of varying the TVSEP threshold is that as we increase the threshold, the number of filters whose $\text{MinTVS}$ value exceeds the threshold reduces. Thus, as we reduce $\eta$, we increase the acceptable degree of TV-separation, thus making the pruning more gentle. We state the algorithm formally in Algorithm 5.

## 7 EXPERIMENTAL RESULTS

In this section, we detail our experimental results. We aim to compare our method with various existing structured pruning techniques without any fine-tuning. Our experiments utilize the standard CIFAR-10 (Krizhevsky et al., 2009) and Imagenet (Russakovsky et al., 2015) datasets under the MIT license. For computing moments, we partition the test set, which contains 10,000 samples, into two subsets of size 7,500 and 2,500 in the case of CIFAR10; for Imagenet, we partition the validation set into two subsets of size 40000 and 10000. We use the larger subset for the computation of moments, and the smaller subset for testing the accuracy of the pruned model. For our hardware setup, we refer readers to section E.

We aim to answer three broad questions with our slate of experiments. First, do well trained CNNs have more filters with well-separated filter outputs than models that are not well trained? Second, using the LDIFF mechanism that utilizes the separability of intermediate features, can we gauge the "difficulty" of pruning a given layer? And last, how does our pruning mechanism compare with other existing methods in the scenario where training data and the loss function are unavailable?

For further experimental results, we refer the reader to the supplementary material.

### 7.1 VALIDATING THE DISCRIMINATIVE FILTERS HYPOTHESIS: WELL-TRAINED MODELS HAVE FILTERS WITH WELL SEPARATED OUTPUTS

In this set of experiments, we aim to show that models that generalize well have filters with well-separated output features, whereas models that generalize poorly do not. To measure separation, we compute

$$\overline{\text{MinTVS}}(l) = \frac{1}{N_{out}^L} \sum_{j=1}^{N_{out}^L} \text{MinTVS}(l,j),$$

that is, the mean minimum classwise TV-separation of filters in each layer. In order to obtain models that do not generalize well, we modify the weights of a model with high test accuracy by adding zero-mean gaussian noise to the weights. In our experiments, we progressively increase the variance of the gaussian noise, in order to increase the test error. Furthermore, measure $\overline{\text{MinTVS}}(l)$ values for untrained models for additional comparison. Our experiments focus on VGG16 and ResNet18 models trained on the CIFAR10 and Imagenet datasets. For both architectures, and for both datasets, we observe that as the variance of the noise increases (and thus, the test error decreases), the average TV-separation per layer decreases. We also observe that the average TV-separation for untrained models remains low, and almost constant. Curiously, we note that even with the addition of Gaussian noise, the separation does not significantly change in the layers close to the input, particularly for VGG models. This therefore supports our hypothesis that models which generalize well possess discriminative filters, whereas models that do not generalize well do not.

### 7.2 DETERMINING WHICH LAYERS ARE DIFFICULT TO PRUNE WITH LDIFF

In this section, we investigate the utility of the LDIFF score for determining which layers can be extensively pruned, and which can't. The experiment we conduct is as follows. First, we compute the

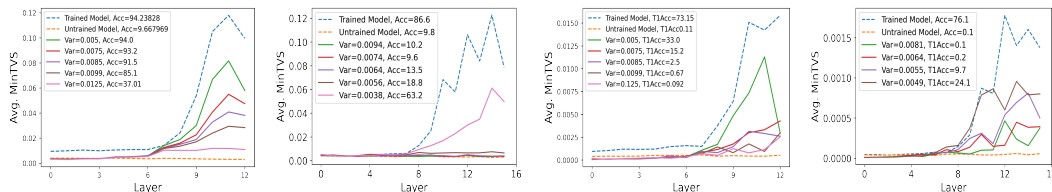

(a) VGG16 trained on CIFAR10 with Gaussian Noise

(b) ResNet18 trained on CIFAR10 with Gaussian Noise

(c) VGG16 trained on Imagenet with Gaussian Noise

(d) ResNet18 trained on Imagenet with Gaussian Noise

Figure 1: Comparison of accuracies of models with weights perturbed by Gaussian noise with fully trained and untrained models.

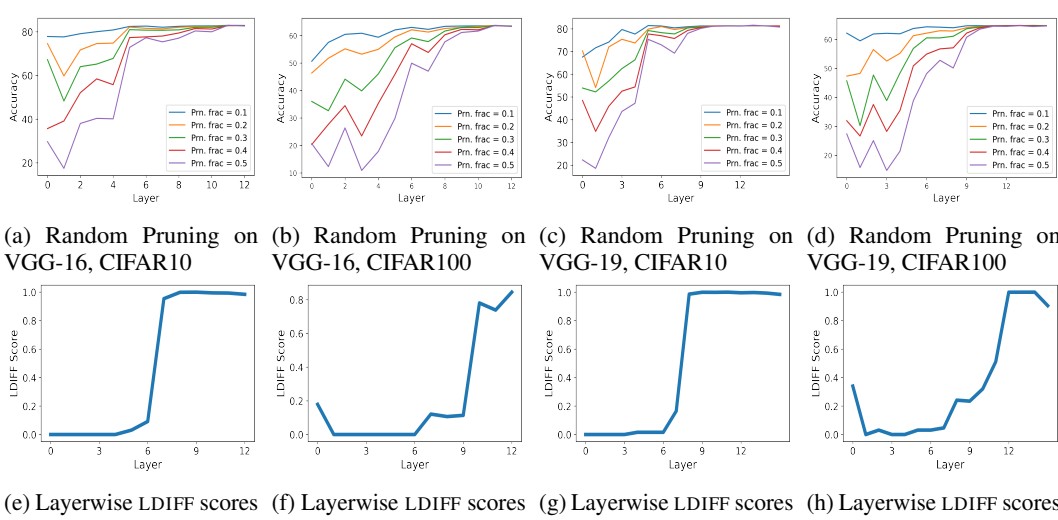

(a) Random Pruning on VGG-16, CIFAR10

(b) Random Pruning on VGG-16, CIFAR100

(c) Random Pruning on VGG-19, CIFAR10

(d) Random Pruning on VGG-19, CIFAR100

(e) Layerwise LDIFF scores for VGG-16, CIFAR10

(f) Layerwise LDIFF scores for VGG-16, CIFAR100

(g) Layerwise LDIFF scores for VGG19, CIFAR10

(h) Layerwise LDIFF scores for VGG19, CIFAR100

Figure 2: Top row: Effect of uniform random unstructured pruning of single layers on VGG16/19 models trained on CIFAR10/100. 'Prn_frac' refers to the percent of the weights removed from the layer. Bottom row: LDIFF scores for corresponding unpruned models.

LDIFF scores for the VGG16 -19 models trained on the CIFAR10 and CIFAR100 datasets. We use 512 samples for CIFAR10, sampled from the appropriate partition of the test set, and 4096 samples for CIFAR100. Then, we prune weights randomly from each layer in isolation; that is, for each experiment, we only prune weights from a single layer. We then measure the test accuracies. We observe that the LDIFF scores are small for the initial layers, thus indicating that those layers cannot be randomly pruned. This is borne out by our experimental results for both models trained on the CIFAR10 and the CIFAR100 dataset. Last, we infer, based on the LDIFF scores and test accuracies of randomly pruned models, that pruning ratios of $\gamma$LDIFF$(l, \eta)$, where $\gamma < 1/2$ would result in negligible loss of accuracy. Collectively, these observations support the use of the LDIFF score for determining the extent to which a layer can be pruned. These results are displayed in 5. Further discussion and experiments are provided in the supplemental material in C.

## 7.3 EFFECTIVENESS OF ITERTVSPRUNE FOR STRUCTURED PRUNING WITHOUT FINE-TUNING

In this section, we compare the ITERTVSPRUNE algorithm with some existing baselines, in the setting where we do not have access to the training set or loss function. We consider VGG16, and -19 models trained on CIFAR10, and ResNet50 and -56 models trained on Imagenet and CIFAR10 respectively. In this set of experiments, we run ITERTVSPRUNE to solve equation P for each model and dataset; then, having obtained the pruned model satisfying the accuracy constraint, we compare with three baselines, using the same sparsity pattern as obtained by ITERTVSPRUNE. The baselines

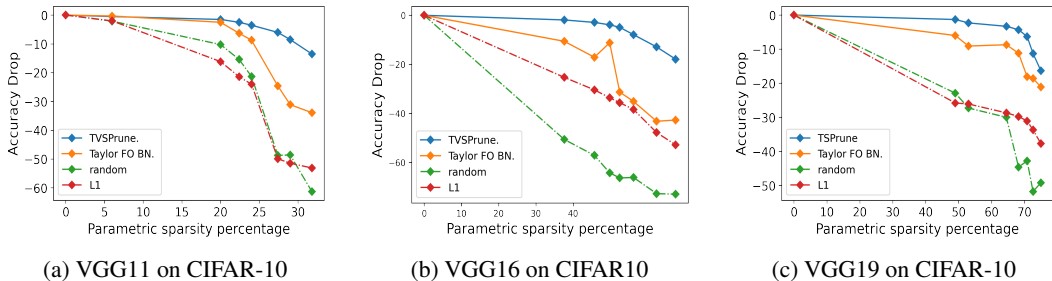

|                              |                              |                              |
|:----------------------------:|:----------------------------:|:----------------------------:|
| (a) VGG11 on CIFAR-10        | (b) VGG16 on CIFAR10         | (c) VGG19 on CIFAR-10        |

Figure 3: Comparison of accuracies of ITERTVSPRUNE  on VGG models trained on CIFAR10

Table 1: Pruning ResNet Models with no training dataset and no finetuning on CIFAR-10 and Imagenet

| Model | Dataset | Param. Sparsity | ITERTVSPRUNE | CHIP | L1 | Rand.-3 |
|---|---|---|---|---|---|---|
| ResNet50 | ImageNet | 4.76% | **-3.02%** | -3.41% | -7.86% | -13.9% |
| | | 9.98% | -10.21% | **-10.08%** | -48.2% | 43.1% |
| | | 24.65% | -31.3% | -34.2% | - | - |
| ResNet56 | CIFAR10 | 3.5% | -1.47% | **-1.36%** | -5.46% | -7.12% |
| | | 7.6% | **-4.82%** | -5.56% | N/A | -9.41% |
| | | 12.3% | **-9.86%** | -10.22% | -17.41% | -21.30% |

chosen were $L_1$ based pruning, CHIP (Sui et al., 2021) (for ResNet models) and a first order gradient based score based on (Prakash et al., 2019; Molchanov et al., 2019a) (for VGG nets), and uniform random pruning (accuracy is the average of three trials of random pruning); all were run without fine tuning. Modifications to implementations are discussed in Appendix E. We report our results for VGG nets on CIFAR10 in Figure 7, and for ResNet models in Table 1. We observe that our method consistently outperforms the baselines in the task of structured pruning of models without fine tuning the model with the training set. Notably, for VGG19 trained on CIFAR10, we are able to remove more than 60% of parameters with minimal loss in accuracy, far exceeding the nearest baseline.

## 8   CONCLUSIONS AND DISCUSSION

In this work, we propose a new paradigm for pruning, which we call distributional pruning, which only requires access to the data distribution. We make the observation that models with high predictive performance possess a mixture of filters that discriminate well, and those that discriminate poorly. Motivated by these observations, we argue that we can prune non-discriminative filters to avoid significant loss of test error. We use the TV Distance to quantify the discriminative ability of filters, using which we define the heuristic LDIFF score and derive the TVSPRUNE Algorithm.

The TVSPRUNE algorithm's drawbacks are as follows. First, since classwise distances must be computed, the number of separations for each filter that the algorithm needs to estimate are quadratic in the number of classes; limiting the scalability of the algorithm. Next, the relationship between TV-separability and generalization in DNNs is not fully understood, and requires further investigation. Future research directions include (a) improving the scalability of this algorithm to massive datasets, (b) evaluating other measures of similarity between distributions, and (c) analyzing parameter quantization through the lens of distributional separability.

## ACKNOWLEDGEMENTS

We authors gratefully acknowledge AMD for their support. The authors also thank Ramaswamy Govindarajan (Professor, IISc), Himanshu Jain (IISc), Raghavendra Prakash (AMD), and Ramasamy Chandra Kumar (AMD) for their insight and assistance in this work.

The authors thank the reviewers for their valuable feedback which has helped us improve our work.

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

APPENDIX

In this supplemental material, we provide additional detail and empirical results to support the claims of the main paper. We organize this document as follows.

1. In section A, for the sake of comprehensiveness, we provide further discussion of relevant literature, including a discussion on the discriminative ability of convolutional filters.

2. In section B, we present additional discussion required for clarification purposes, including a brief discussion adopting the machinery used in this paper to feedforward models.

3. In section C, we provide further details regarding the derivation of the LDIFF scores.

4. In section E, we provide additional experiments in using the TVSPRUNE and TVSPRUNE algorithms for structured pruning. We provide plots showing the number of filters pruned in each layer for the same experiments presented in Section 7.3 of the original paper.

5. In section D we state a simple variation of the ITERTVSPRUNE algorithm that aims to solve a slight variation of the pruning problem.

   In the sequel, we list additional material added to this document.

6. In section E, we provide additional experiments illustrating the utility and effectiveness of the ITERTVSPRUNE algorithm.

## A  ADDITIONAL LITERATURE SURVEY

In this section, we survey contemporary and historic results in the area of neural network pruning. We focus on filter pruning in this work, but there exists a wide variety of works addressing problem of unstructured, or weight pruning. For a more detailed discussion on relevant results in unstructured pruning, refer to (Hoefler et al., 2021; Blalock et al., 2020).

**Filter Pruning**   Filter pruning refers to processes by which entire filters or channels are pruned from a model; this can be further extended to individual filters(Zhong et al., 2021) or 'stripes' (Meng et al., 2020) within filters. Initial results on structured pruning removed filters based on the $L1$ norm of the weight tensors (Li et al., 2016). More recent advances involve using gradient based scores (Molchanov et al., 2019b; Liu et al., 2021b; Molchanov et al., 2019a), rank minimization (Lin et al., 2020a), and linear replaceability and channel independence (Joo et al., 2021; Sui et al., 2021), and other geometric properties (He et al., 2018; 2019; Li et al., 2020; Lin et al., 2020b; Yu et al., 2018). Other works, such as (Liebenwein et al., 2019; Baykal et al., 2018) developed sampling based methods for structured pruning that also provided generalization bounds. Lastly, in (Zhuang et al., 2018; Liu et al., 2021a), the discriminative capacity of filters is used to inform the pruning strategy by introducing discrimination-aware loss functions. Our work differs from these in two key respects - first, we rank filters based on the discriminative ability of their outputs, and second, we use a novel metric based on the discriminative ability of filters, based on classwise distances between class-conditional output distributions of filters, to decide which layers can be pruned effectively.

**Data-Free Pruning**   Data free pruning has recently gained significant interest (Hoefler et al., 2021); however, while there are a variety of methods that do not use data for the task of pruning, there are few methods that do not require fine tuning or retraining. Early works in this regime include (Srinivas & Babu, 2015), which measured similarity between neurons, and merged similar neurons together. More recently, (Tanaka et al., 2020) proposed the SynFlow method, an unstructured, data free pruning method that relied on preserving gradient flow. This work was expanded upon in (Gebhart et al., 2021; Patil & Dovrolis, 2021), which utilize the Neural Tangent Kernel to improve upon SynFlow. Other approaches include (Mussay et al., 2021), which uses coresets of weights to perform structured pruning. Our work differs from this since we utilize the class conditional distributions of the filter outputs to rank the filters, which we either obtain *a priori*, or compute using previously unseen data. In this section, we briefly survey works pertaining to discriminative ability of neural networks, and unstructured pruning. We do so for the sake of comprehensiveness, and to discuss works that are tangentially relevant to ours.

**Discriminative Ability of Convolutional Neural Networks**    Studying the discriminative ability of features and representation at either the layer level or the filter level is an active area of research, with applications in speech identification (Yadav & Rai, 2018), scene classification (Zuo et al., 2014), face recognition (Wen et al., 2016), and domain adaptation (Lee et al., 2019). Understanding the discriminative capacity of features is also crucial toward understanding adversarial robustness (Jetley et al., 2018; Ilyas et al., 2019) and model interpretability (Bau et al., 2017). In (Wong et al., 2021), pretrained set of convolutional layers is given as the input to a linear classifier which is trained with a sparsity-inducing regularizer to identify discriminative filters. In (Ortiz-Jimenez et al., 2020), the discriminative capacity of intermediate representations is used to analyze decision boundaries of CNNs. However, to the best of our knowledge, there are no works that utilize the discriminative capacity of filters for pruning models.

**Discriminative Filters in Pruning**    Discriminative filters have also been applied to structured pruning of neural networks. In (Zhuang et al., 2018; Liu et al., 2021a), the discriminative capacity of filters is used to inform the pruning strategy. In particular, the authors define the discriminative ability of a filter as the impact of it's ouput on the final classifer, and introduce a discrimination aware loss function, with which the model is fine tuned. Once this is done, the authors propose a greedy algorithm for channel selection that makes use of this loss function and the associated gradients. This method differs from this work, as we explicitly define discriminative ability in terms of the TV separation, and not on the impact on the final classifier.

**Unstructured Pruning**    In this section, we provide a brief overview of unstructured pruning techniques. While unstructured pruning is not the focus of this work, we provide this summary in the interest of completeness. and Unstructured pruning refers to processes by which individual weights are pruned from a model (Hoefler et al., 2021). Early works on unstructured pruning include (LeCun et al., 1989; Hassibi & Stork, 1992), which relied on diagonal approximations of the Hessian of the loss function. However, such methods are not scalable to very large models, prompting the creation of unstructured pruning methods that married a variety of techniques to achieve dramatic parametric sparsity (Han et al., 2015; Iandola et al., 2016). More recently, (Frankle & Carbin, 2018) proposed the Lottery Ticket Hypothesis, which asserts that networks contain subnetworks which, when trained from scratch, achieve similar accuracies to the original model; stronger versions of this result were proved in (Malach et al., 2020; Pensia et al., 2020). This remains an active area of research (Blalock et al., 2020).

# B    Additional Discussion

In this section, we include additional discussion on certain aspects of this work. In particular, we briefly discuss the extension of our method to linear, feedforward networks, and provide additional discussion on the definitions and use of patches.

## B.1    Extension of proposed methodology to linear layers

Our proposed methodology can easily be extended to feedforward/linear layers as well. Broadly speaking, we consider the layerwise outputs to be the outputs of linear layers, as opposed to convolutional layers. In that case, it is possible to derive the same measures of distributional discrepancy. We provide a more rigorous derivation below:

Let $\Phi^{l-1}(X)$ be the output of the $l-1$-th linear layer, and let $W^l$ be the weight tensor of the $l$-th layer. Then,

$$Y_l(X) = \langle \Phi^{l-1}(X), W^l \rangle. \tag{7}$$

If $Y_l(X)$ is a vector, let $Y_l^j(X)$ be the $j$th element (the $j$th neuron). Then, we simply define

$$\bar{Y}^\alpha = \mathbb{E}_{(X,u)\sim\mathcal{D}^\alpha}\left[Y(X)\right] \text{ and } \sigma^2_{l,j,\alpha} = \bar{\mathbb{E}}_{(X,u)\sim\mathcal{D}^\alpha}\left[\|Y(X) - \bar{Y}^\alpha\|^2\right] \tag{8}$$

as we did for convolutional layers. However, in this case, outputs are vectors. Thus, each $Y_l^j$ is a single real number (along with the expectations and the variances). Using this, we can then define

$$\texttt{HELLD}^2\left(\mathbb{Q}_1, \mathbb{Q}_2\right) = 1 - \left(\frac{2\sigma_1\sigma_2}{\sigma_1^2 + \sigma_2^2}\right)^{\frac{d}{2}} e^{-\frac{\Delta}{4}}, \ \Delta = \frac{(\mu_1 - \mu_2)^2}{\sigma_1^2 + \sigma_2^2} \tag{9}$$

and

$$\text{MinTVS}(l,j) = \min_{\alpha,\beta} \ \text{TV}\left(\mathcal{D}^\alpha_{l,j}, \mathcal{D}^\beta_{l,j}\right) \geq \min_{\alpha,\beta} 1 - \exp(-\Delta^{\alpha,\beta}_{l,j}/4) \tag{10}$$

where
$$\Delta_{l,j}^{\alpha,\beta} = (\bar{Y}_{l,j}^{\alpha} - \bar{Y}_{l,j}^{\beta})^2 \left( \sigma_{l,j,\alpha}^2 + \sigma_{l,j,\beta}^2 \right)^{-1}. \tag{11}$$

## B.2 DESCRIPTION OF PATCHES

A patch is a subset of the input datum with which a dot product with the convolutional filter is taken. The input datum is made up of several patches which may or may not overlap. We describe this formally in the sequel.

For simplicity, assume that the input is a in $\mathbb{R}^{M \times M}$, and each convolutional filter is a tensor in $\mathbb{R}^{K \times K}$ where $K < M$. In a convolution operation, the filter is applied to subsets of size $K \times K$ of the input; that is, the output is a dot product of the filter and that subset of elements of the input. Each such subset is called a **patch**. We describe this further in Section H of the supplemental material. Furthermore, we refer the reader to, for example, (Goodfellow et al., 2016).

## C THE LDIFF SCORE: ADDITIONAL CONTEXT AND EMPIRICAL ANALYSIS

In this section, we provide additional context and information regarding the LDIFF score described in Section 6.1, which is a heuristic used to quantify the extent to which a layer may be randomly pruned.

### C.1 REFINEMENTS OF THE LDIFF SCORE

In this section, we discuss refinements of the LDIFF score presented in section 5. Recall that in this work, we focus on identifying and pruning non-discriminative filters from individual layers. Thus, as mentioned previously, our goal is to derive a score, leveraging the aforementioned hypothesis and using the discriminative ability of the filters in a given layer, that we can use to deduce whether or not a layer is difficult to prune.

Assuming the hypothesis is true, a good measure of the extent to which a model can be pruned could be given by the fraction of filters that are highly discriminative. Thus, if a layer has only a few highly discriminative filters, it should be possible to sparsify that layer extensively.

In the LDIFF score defined in equation 6, we fix the threshold to identify discriminative filters. We first present methods with fixed thresholds that utilize statistics of the layerwise MinTVS scores here.

#### C.1.1 MEDIAN-BASED LDIFF SCORES WITH A FIXED THRESHOLD

We begin by recalling definitions from the original paper.

Let $\widehat{\text{MinTVS}}(l)$ be the median value of $\text{MinTVS}(l, j)$ for a given $l$ (that is, with respect to $j$). We recall that a filter is discriminative if it is $\text{TVSEP}(\eta)$ for some $\eta > 0$; furthermore, recall that $\eta$ can vary from model to model, and dataset to dataset. With this, we first define the following sets.

- Suppose that for at least one $j$, the corresponding filter is $\text{TVSEP}(\eta)$ for some $\eta > 0$. We define the sets
$$S_{\eta,l} = \{j : \Delta_{l,j}^* < \eta\}, \text{ and } \hat{S}_l = \{j : \Delta_{l,j}^* \geq \hat{\Delta}_l.\}$$
- The set $S_{\eta,l}$ captures the number of filters that are not $\text{TVSEP}(\eta)$, and thus non-discriminative.
- The set $\hat{S}_l$ is the set of $\Delta_{l,j}^*$ greater than the median.

Let $|S|$ denote the cardinality of a set $S$. We then define
$$\text{LDIFF}_0(l) = 1 - \frac{|S_{\eta,l} \cap \hat{S}_l|}{|\hat{S}_l|}. \tag{12}$$

The set $S_{\eta,l} \cap \hat{S}_l$ captures the number of non-discriminative filters in the top two quantiles. Thus, $\text{LDIFF}_0(l)$ captures the fraction of filters in the top two quantiles that are discriminative. If this quantity equals 1, we conclude that at least half the filters are $\text{TVSEP}(\eta)$; thus $\text{LDIFF}_0$ captures the mix between discriminative and non-discriminative filters. Thus, if $\text{LDIFF}_0(l)$ is close to 1, then the layer is difficult to prune.

This is illustrated in Figure 9, where we plot the $\text{LDIFF}_0(l)$ scores for each layer for VGG16 and VGG19 models trained on the CIFAR10 and CIFAR100 datasets. We choose $\eta = 0.025$, and we see that the initial layers of VGG16 and VGG19 models trained on CIFAR10 appear to be very easy to prune, since there should be no discriminative filters. Instead, however, those layers cannot be effectively sparsified.

In order to compensate for this, we chose another way to quantify the mix between discriminative and non-discriminative filters in a layer. Specifically, we define LDIFF as in equation 6, which captures the ratio of the separations that are below the median $\hat{\Delta}_l$, and those above it. As we will show in the sequel, this is a more effective variant for capturing the difficulty of pruning filters, or indeed weights, from a layer.

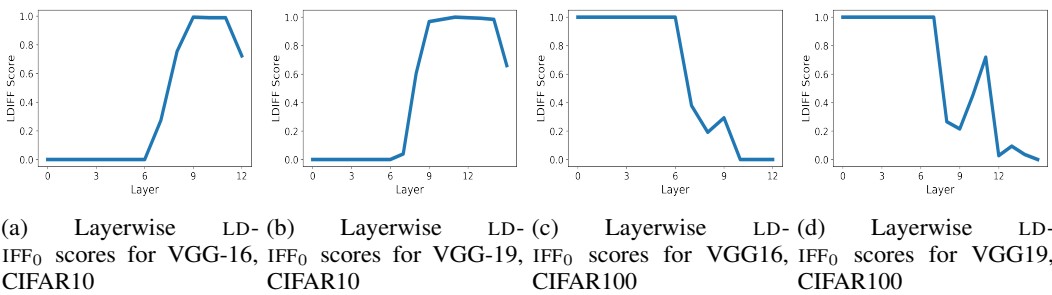

(a) Layerwise LD-IFF$_0$ scores for VGG-16, CIFAR10

(b) Layerwise LD-IFF$_0$ scores for VGG-19, CIFAR10

(c) Layerwise LD-IFF$_0$ scores for VGG16, CIFAR100

(d) Layerwise LD-IFF$_0$ scores for VGG19, CIFAR100

Here, we observe that the $\text{LDIFF}_0$ score is effective at determining whether a layer is difficult to prune for models trained on CIFAR100, but is seemingly not as effective for models trained on CIFAR10. In particular, for models trained on CIFAR10, we observe that the initial layers receive low scores - since filter outputs for those models are not well separated. This lends credence to the notion that if there are no filters in a layer that are $\text{TVSEP}(\eta)$, then we should avoid pruning that layer entirely. In the sequel, we discuss variants of the scores defined herein and propose variants, including the score proposed in the final paper.

### C.2 FLEXIBLE THRESHOLD LDIFF SCORES

A potential drawback of using fixed thresholds for discriminative ability, however, is that the scale of the distributional separation may vary from layer to layer. That is a value of $\eta$ that ensures discriminative ability in one layer may not apply to another. In this section, we extend this definition to a more flexible variant, wherein we utilize layerwise thresholds. We consider multiple variants, including cases where thresholds are fixed a priori and variants where thresholds are fixed based on statistics of the $\text{MinTVS}$ values for a given layer. Furthermore, layerwise thresholds for discriminative ability provide the added benefit of being able to capture the 'mix' of discriminative and non-discriminative filters.

### C.2.1 COUNTING-BASED LDIFFSCORES USING LAYERWISE MinTVS STATISTICS

In this section, we focus on methods that utilize the statistics of $\text{MinTVS}$ values for a given layer to determine pruning thresholds. A variant of the method used in the main paper is as follows.

Define $\overline{\text{MinTVS}}(l)$ to be the mean $\text{MinTVS}$ for a given $l$ (that is, with respect to $j$), and let $\gamma > 0$ be a constant. Then, define

$$\overline{\text{LDIFF}}_C(l) = \frac{\sum \mathbf{1}\{\text{MinTVS}(l, j) < \gamma \overline{\text{MinTVS}}(l)\}}{\sum \mathbf{1}\{\text{MinTVS}(l, j) > \gamma \overline{\text{MinTVS}}(l)\}} \tag{13}$$

The $\overline{\text{LDIFF}}_C(l)$ score utilizes the mean $\text{MinTVS}$ score for a layer to capture the *relative* discriminative ability of filters in a layer, along with a constant $\gamma_l$. For each layer, this measure counts the number of filters that are discriminative *relative* to the $\text{MinTVS}$ score for that layer. As mentioned previously, useful thresholds of discriminative ability can vary significantly from architecture to architecture, layer to layer, and dataset to dataset.

### C.2.2 MASS-BASED LDIFFSCORES USING LAYERWISE MinTVS STATISTICS

The previous counting-based measures of pruneability, even with flexible thresholds, fail to account for the wide range of values that MinTVS scores can take, particularly in wide networks with a small number of very discriminative filters. In order to address this issue, we define the following variant of the LDIFF score.

$$\overline{\text{LDIFF}}(l) = \frac{\sum \text{MinTVS}(l,j)1\{\text{MinTVS}(l,j) < \gamma\overline{\text{MinTVS}}(l)\}}{\sum \text{MinTVS}(l,j)\mathbf{1}\{\text{MinTVS}(l,j) > \gamma\overline{\text{MinTVS}}(l)\}} \quad (14)$$

This variant of LDIFF score compares the ratio of $\text{MinTVS}(l,j)$ scores that lie below $\gamma\overline{\text{MinTVS}}(l)$ to the sum of $\text{MinTVS}(l,j)$ scores that lie above $\gamma\overline{\text{MinTVS}}(l)$. Thus, we compare the 'mass' of the discriminative ability of scores below the (variable) threshold to those above it. Hence, we use the term "Mass-based" LDIFF scores. This captures the spread of discriminative ability, and can be shown to more effectively predict which layers can be randomly pruned.

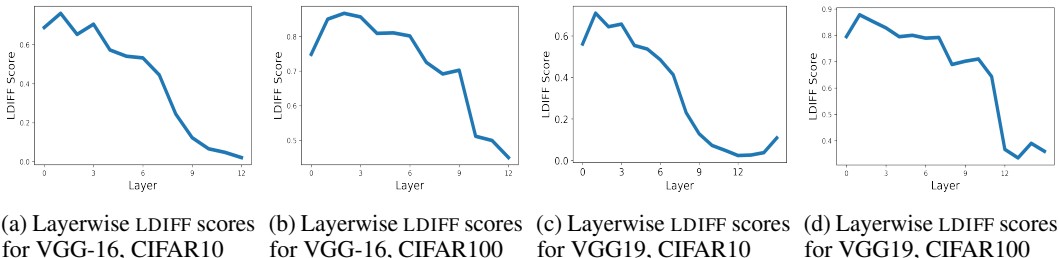

(a) Layerwise LDIFF scores for VGG-16, CIFAR10

(b) Layerwise LDIFF scores for VGG-16, CIFAR100

(c) Layerwise LDIFF scores for VGG19, CIFAR10

(d) Layerwise LDIFF scores for VGG19, CIFAR100

Figure 5: Mass based LDIFF scores. Here, a higher score means it is more difficult to prune.

## D VARIANTS OF ITERTVSPRUNE

In this section, we describe variants of the TVSPRUNE algorithm. First, we describe a method that utilizes flexible thresholds, and second, we describe a variant to solve the pruning problem with a fixed sparsity budget.

### D.1 TVSPRUNE WITH FLEXIBLE THRESHOLDS

. This section describes a variant of the TVSPRUNE and ITERTVSPRUNE algorithms that utilize variable thresholds for discriminative ability. In this case, we specifically use the mean MinTVS score to determine layerwise pruning thresholds. Furthermore, we use the mass-based LDIFF score described in equation 14 to determine which layers to prune.

**Algorithm 3:** TVSPRUNE$_{\text{mean}}$

**Input:** Dataset $\hat{\mathcal{D}} = \{X_i, u_i\}_{i=1}^{M}$, Pretrained CNN with parameters $\mathcal{W} = (W_1, \cdots, W_L), \gamma > 0$, layer difficulty threshold $\nu$

Compute $\bar{Y}_j^{l,\alpha}, \sigma_{l,j,\alpha}^2$ for all $l, j,$ and $\alpha$.

Compute $\Delta_{l,j}^{\alpha,\beta}$ for each $l, j, \alpha, \beta$.

**for** $l \in [L]$ **do**

    Compute MinTVS$(l, j)$ for all $l, j$

    Compute $\overline{\text{MinTVS}}(l) =$

    $\frac{1}{N_{out}^l} \sum_{j=1}^{N_{out}^l} \text{MinTVS}(l, j)$

    Compute $\overline{\text{LDIFF}}(l)$ from equation 14

    **if** LDIFF$(l) < \nu$ **then**

        **for** $j \in [N_{out}^l]$ **do**

            **if** MinTVS$(l, j) < \gamma \overline{\text{MinTVS}}(l)$ **then**

                $W_j^l \leftarrow \mathbf{0}$

**Output:** $\hat{\mathcal{W}} = (\hat{W}_1, \cdots, \hat{W}_L)$, where $\text{supp}(\hat{W}_l) \leq \text{supp}(W_l) \; \forall \; l$

**return** $\hat{\mathcal{W}}$

---

**Algorithm 4:** ITERTVSPRUNE$_{\text{mean}}$

**Input:** Dataset $\hat{\mathcal{D}} = \{X_i, u_i\}_{i=1}^{M}$, Pretrained CNN $\mathcal{W} = (W_1, \cdots, W_L), \gamma_0 > 0$, LDIFF threshold $\nu, \delta_\gamma > 0$, accuracy threshold $t, \gamma_{\min} < \gamma$.

Set $k = 0, \mathcal{W}_{(0)} = \mathcal{W}$

**while** $\gamma \geq \gamma_{\min}$

    $\tilde{\mathcal{W}} = \text{TVSPRUNE}_{\text{mean}}(\mathcal{D}, \mathcal{W}_{(k)}, \gamma, \nu)$

    **if** $f(\tilde{\mathcal{W}}) \leq t$ **then**

        $\mathcal{W}_{(k+1)} \leftarrow \tilde{\mathcal{W}}$

    **else**

        $\gamma \leftarrow \gamma - \delta_\gamma$

        $\mathcal{W}_{(k+1)} \leftarrow \mathcal{W}_{(k)}$

    $k \leftarrow k + 1$

**Output:** $\hat{\mathcal{W}} = (\hat{W}_1, \cdots, \hat{W}_L), f(\hat{\mathcal{W}}) \leq t,$ $\text{supp}(\hat{W}_l) \leq \text{supp}(W_l) \; \forall \; l$

**return** $\hat{\mathcal{W}}$

---

This algorithm functions in a similar fashion to the TVSPRUNE and ITERTVSPRUNE algorithms, except that the thresholds for discriminative ability are flexible (varying between layers), and are decided by the mean of the MinTVS scores for each layer. Furthermore, just as the threshold for discriminative ability were reduced at each iteration of ITERTVSPRUNE, we vary the constant $\gamma$, thus enabling us to flexibly prune each layer even as the distribution of MinTVS scores changes with each pruning iteration.

### D.2 FIXED-BUDGET PRUNING

In this section, we briefly describe a variant of the ITERTVSPRUNE algorithm that, using only the data-generating distribution, prunes a model with a fixed sparsity budget using only the data-generating distribution. This variant attempts to solve the problem of finding the most accurate model subject to a sparsity budget. Formally speaking, this variant of the algorithm attempts to solve

$$\arg\min \{f(\mathcal{W}) \lceil \|\mathcal{W}\|_0 \leq K, \; K > 0\}. \tag{P2}$$

where $\|\mathcal{W}\|_0 = \sum_l \|W_l\|_0$ and where $K$ is a sparsity budget satisfying $K < \|\mathcal{W}\|_0$. While a simple option is to simply increase $\eta$ in the TVSPRUNE algorithm (thus raising the threshold as to which filters are discriminative), the proposed variant makes use of the fact that the distributions of the filter outputs change at each iteration, owing to the filters pruned in the previous step This algorithm,

---

**Algorithm 5:** ITERTVSPRUNE-SB

**Input:** Dataset $\hat{\mathcal{D}} = \{X_i, u_i\}_{i=1}^{M}$, Pretrained CNN $\mathcal{W} = (W_1, \cdots, W_L)$, initial TVSEP threshold $\eta$, LDIFF threshold $\nu, \delta_\eta > 0$, sparsity budget $K, \eta_{\min} < \eta$.

Set $k = 0, \mathcal{W}_{(0)} = \mathcal{W}$

**while** $\eta \geq \eta_{\min}$

    $\tilde{\mathcal{W}} = \text{TVSPRUNE}(\mathcal{D}, \mathcal{W}_{(k)}, \eta, \nu)$

    **if** $\|\tilde{\mathcal{W}}\|_0 \geq K$ **then**

        $\mathcal{W}_{(k+1)} \leftarrow \tilde{\mathcal{W}}$

    **else**

        $\eta \leftarrow \eta - \delta_\eta$

        $\mathcal{W}_{(k+1)} \leftarrow \mathcal{W}_{(k)}$

    $k \leftarrow k + 1$

**Output:** $\hat{\mathcal{W}} = (\hat{W}_1, \cdots, \hat{W}_L), \|\hat{\mathcal{W}}\|_0 \leq K,$

**return** $\hat{\mathcal{W}}$

---

ITERTVSPRUNE-SB (SB refers to sparsity budget), iteratively calls the TVSPRUNE algorithm to

prune a fraction of filters at each step. At the $k$th iteration, if $\|\tilde{\mathcal{W}}\|_0 < K$; that is, the fraction of parameters drops below the sparsity budget $K$, we rewind to the previous value of $\mathcal{W}_k$, and reduce $\eta$, thereby ensuring that we prune more gently. The drawback of using this algorithm is that $\eta_{\min}$ needs to be chosen carefully so that we are guaranteed to prune some filters at each iteration. While we do not directly control for accuracy, we rely on our hypothesis that we may remove non-discriminative filters without suffering catastrophic losses in accuracy.

## E    ADDITIONAL EXPERIMENTS USING ITERTVSPRUNE

In this section, we provide additional experimental details on the use of the ITERTVSPRUNE algorithm.

### HARDWARE SETUP

Our experiments were conducted on two systems, one with an NVIDIA GTX1060 GPU and an Intel i7-7700, and one with dual NVIDIA RTX3090 GPUs and an Intel i9-11900F. All experiments are done using the PyTorch framework, in particular, for obtaining the output features for each filter. Moments and TVSEPvalues were computed using standard numpy packages.

### DATA SET SPLITS

We briefly discuss the split of the data used in our experiments. As mentioned in the main paper, we do not make use of the training data at all. However, since we are unable to obtain sufficient data that would be drawn from the class conditional distributions of the CIFAR10 and CIFAR100 datasets (i.e. obtaining a large number of images of cats, planes, horses etc), we utilize the test data. We partition the test data into two, with one partition receiving 7500 images, and the other 2500. We describe this in Table 2.

Table 2: Breakdown of dataset splits used in our experiments.

| Dataset | Training Set | Mean Computation Set | Test Set |
|---------|--------------|----------------------|----------|
| CIFAR10 | Not used | 7500 images from test set | 2500 images from Test set |
| CIFAR100 | Not used | 7500 images from test set | 2500 images from Test set |
| Imagenet | Not used | 40000 images from validation set | 10000 images from validation set |

**Hyperparameters**    For all experiments, the value of $\eta$ as used in Algorithm 2 of the main manuscript is chosen empirically, and varies for each experiment. The value for $\tilde{\delta}$ is chosen to be 0.15. The value of $\nu$ is chosen to be 0.5. We state the hyperparameters used in or experiments using fine-tuning in Table 3.

Table 3: Hyperparameters chosen for fine-tuning pruning model with the CIFAR10 and Imagenet datasets.

| Hyperparameter | Learning Rate | Batch Size | Epochs | Momentum | Scheduler | Weight Decay |
|----------------|---------------|------------|--------|----------|-----------|--------------|
| **Value/CIFAR10** | 0.001 | 128 | 50 | 0.05 | Cosine | 0.9 |
| **Value/Imagenet** | 0.001 | 256 | 70 | $10^{-6}$ | Cosine | .0.99 |

### BASELINE METHODS

In this section, we briefly describe our modifications to common baselines.

1. CHIP Sui et al. (2021): We ensure that the algorithm only has access to the validation set, which is split into a 40000/10000 pair. When running the algorithm, we limit the number of samples seen by the algorithm to 512 (1 batch of 512) for CIFAR10. For Imagenet results, we reduce the number of repetitions to 1.

2. Taylor first order methods: we produce a bespoke implementation of the methods described in Molchanov et al. (2019a;b). We allow the algorithm to see 512 samples for CIFAR 10 to estimate gradients, before pruning. We treat this as a one-shot pruning method.

3. Other methods: for L1 and random pruning, we use native PyTorch implementations.

### E.1 Summary of Experimental Results and Methodology

In this section, we provide tables summarizing our empirical results.

We apply ITERTVSPRUNE to achieve accuracy drops of at most 1.5%, 5%, and 10%, on a variety of architectures and datasets. We then check the accuracy observed when we prune the same networks, on the same datasets, with baseline pruning algorithms. We detail this list below. All experiments do not make use of the training set, except experiments where we apply CHIP Sui et al. (2021) toward pruning the models.

1. **ResNet18/CIFAR10**
   **Baselines:** L1 Pruning, Random Pruning, Taylor First Order Pruning
2. **ResNet50/CIFAR10**
   **Baselines:** L1 Pruning, Random Pruning, Taylor First Order
3. **ResNet56/CIFAR10**
   **Baselines:** L1 Pruning, Random Pruning, Taylor First Order, CHIP Sui et al. (2021)
4. **VGG16/CIFAR10**
   **Baselines:** L1 Pruning, Random Pruning, Taylor First Order, CHIP Sui et al. (2021)

### E.2 CIFAR10 Results

In this subsection, we detail our experiments on models trained using the CIFAR10 dataset.

#### E.2.1 VGG Models

In this subsection, we compare VGG models trained on the CIFAR10 dataset. Specifically, we compare the efficacy of ITERTVSPRUNE on VGG11, VGG16 and VGG19 models, with standard baselines. In our experiment, we first use ITERTVSPRUNE$_{\text{mean}}$ to prune the models to a given accuracy threshold; that is, we solve equation P for different values of $t$. Then, using the same sparsity patterns, we apply $L_1$ pruning, random pruning, and first-order gradient-based pruning respectively. Note that here too, we do not fine tune the models after pruning. We plot these results in Figure 6. For clarity, we also present these results in Table 4.

**Parametric sparsity** We observe that using ITERTVSPRUNE dramatically outperforms several standard baselines. Indeed, we are able to prune more than 60% of parameters in VGG19 trained on the CIFAR10 dataset with only a 2% drop in accuracy. We also observe, from table 4, that on VGG16, ITERTVSPRUNE also outperforms contemporary baselines such as CHIPSui et al. (2021).

**Layerwise Structured Sparsity** In Section 7.3, we report the parametric sparsity - that is, the number of parameters pruned from the model using our proposed algorithms. In this section, we consider the layerwise structured sparsity achieved using the ITERTVSPRUNE algorithm. In our experiments, we observe that irrespective of accuracy threshold, most of the pruned filters lie in the final layers of the VGG models, and almost no filters are pruned in the initial layers.

**Key Takeaways** The key takeaway of this slate of experiments is that VGG models trained on CIFAR10 can be sparsified extensively with ITERTVSPRUNE. We see that our method consistently outperforms the closest baselines, often by double digit percentages (in terms of test accuracy) for the same sparsity.

#### E.2.2 ResNet Experiments

In this section, we detail our experiments with ResNet models trained on the CIFAR10 dataset. The models we consider are the ResNet18, -20, -50, and -56 models. Our procedure mirrors that used

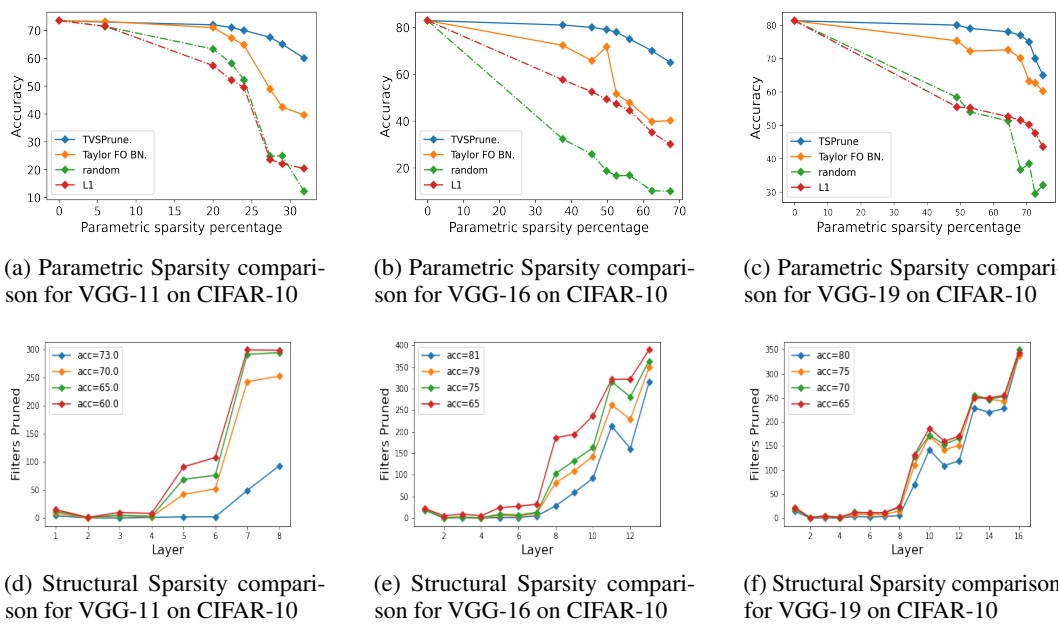

(a) Parametric Sparsity comparison for VGG-11 on CIFAR-10

(b) Parametric Sparsity comparison for VGG-16 on CIFAR-10

(c) Parametric Sparsity comparison for VGG-19 on CIFAR-10

(d) Structural Sparsity comparison for VGG-11 on CIFAR-10

(e) Structural Sparsity comparison for VGG-16 on CIFAR-10

(f) Structural Sparsity comparison for VGG-19 on CIFAR-10

Figure 6: Comparison of accuracies of different pruning algorithms applied to VGG models trained on CIFAR-10

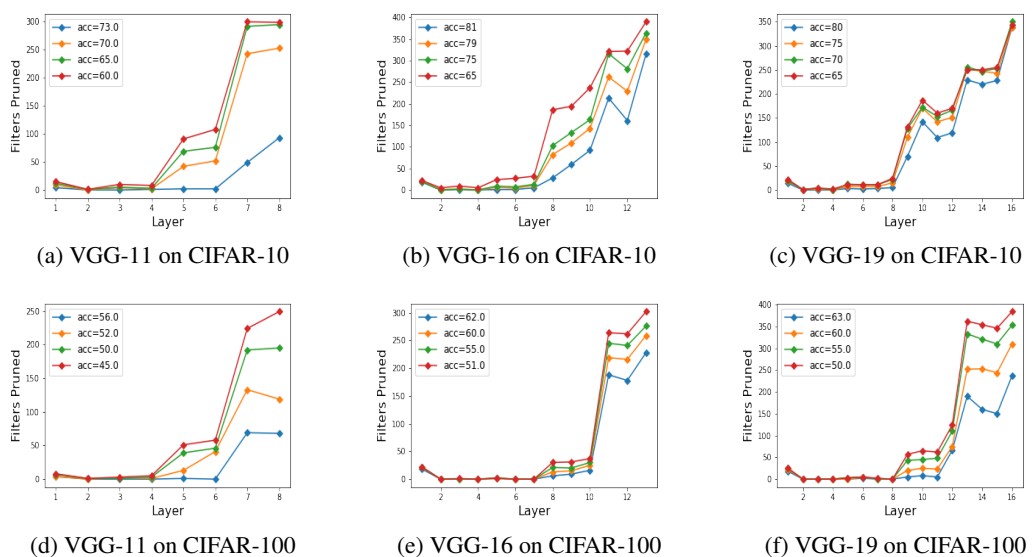

(a) VGG-11 on CIFAR-10

(b) VGG-16 on CIFAR-10

(c) VGG-19 on CIFAR-10

(d) VGG-11 on CIFAR-100

(e) VGG-16 on CIFAR-100

(f) VGG-19 on CIFAR-100

Figure 7: Top row: Structured sparsity pattern after ITERTVSPRUNEfor VGG models trained on CIFAR10. Bottom row: Structured sparsity pattern after ITERTVSPRUNEfor VGG models trained on CIFAR100.

in the previous section, with VGG nets: we use ITERTVSPRUNE to prune models with different accuracy thresholds, and then apply baseline methods with the same sparsity patterns obtained via ITERTVSPRUNE. We present the results of our experiments in Table 5.

**Key takeaways** The key takeaway from this slate of experiments is that ITERTVSPRUNE is effective on residual networks as well. However, it is not as effective on narrow resnet models trained on CIFAR10 as it is on VGG models trained on the same dataset. We observe that the extent to

Table 4: Pruning VGG Models with no training dataset and no finetuning on CIFAR-10

| Model/Param. Sparsity | Pruning Method | Acc. Drop | Comp. with ITERTVSPRUNE |
|---|---|---|---|
| VGG-11/5.99% | ITERTVSPRUNE | 0.52 % | - |
| | Random-3 | 2.12% | -1.6% |
| | $L_1$ | 2.02% | -1.5% |
| | Gradient-FO | **0.32%** | 0.2% |
| | CHIP | - | - |
| VGG-11/24.0% | ITERTVSPRUNE | **3.52 %** | - |
| | Random-3 | 21.32% | -17.8% |
| | $L_1$ | 24.02% | -20.5% |
| | Gradient-FO | 8.72% | -5.2% |
| | CHIP | - | - |
| VGG-11/31.8% | ITERTVSPRUNE | **13.52 %** | - |
| | Random-3 | 61.32% | -47.8% |
| | $L_1$ | 53.12% | -39.6% |
| | Gradient-FO | 33.92% | -20.4% |
| | CHIP | - | - |
| VGG-16/37.6% | ITERTVSPRUNE | **1.9%** | - |
| | Random-3 | 50.6% | -48.7% |
| | $L_1$ | 25.3% | -23.4% |
| | Gradient-FO | 10.6% | -6.5% |
| | CHIP | 4.1% | -2.2% |
| VGG-16/52.5% | ITERTVSPRUNE | **4.9%** | - |
| | Random-3 | 66.3% | -61.4% |
| | $L_1$ | 35.6% | -30.7% |
| | Gradient-FO | 31.3% | -26.4% |
| | CHIP | 7.9% | -3.0% |
| VGG-16/67.5% | ITERTVSPRUNE | **12.9%** | - |
| | Random-3 | 72.7% | -69.8% |
| | $L_1$ | 47.8 | -34.9% |
| | Gradient-FO | 43.2% | -30.3% |
| | CHIP | 16.6% | -3.7% |
| VGG-19/49.0% | ITERTVSPRUNE | **1.3%** | - |
| | Random-3 | 22.9% | -21.6% |
| | $L_1$ | 25.8% | -24.5% |
| | Gradient-FO | 6.0% | -4.7% |
| | CHIP | - | - |
| VGG-19/64.5% | ITERTVSPRUNE | **4.3%** | - |
| | Random-3 | 30.0% | -27.7% |
| | $L_1$ | 28.7% | -24.4% |
| | Gradient-FO | 8.7% | -4.4% |
| | CHIP | - | - |
| VGG-19/72.3% | ITERTVSPRUNE | 70% | - |
| | Random-3 | 51.8% | -40.5% |
| | $L_1$ | 33.7% | -22.4% |
| | Gradient-FO | 18.6% | -7.3% |
| | CHIP | - | - |

which "narrow" ResNet models, such as ResNet20, can be sparsified is significantly less than "wider" ResNet models, such as ResNet18, adapted for CIFAR10.

### E.3 CIFAR100 RESULTS

In this section, we detail our experiments on models trained on the CIFAR100 dataset. The architectures considered were VGG11, VGG16, and VGG19. The experimental setup was identical to the previously presented experiments with the same models trained on the CIFAR10 dataset. We plot those results below in Figure 8, and provided a detailed snapshot in Table 6.

### E.4 EFFECT OF SIZE OF VALIDATION SET

We conduct a brief experiment to illustrate the extent to which the size of the set of samples used to compute the class-conditional moments, and the MinTVS scores, affects the ranking in terms of MinTVS. For a VGG16 model trained on the CIFAR10 dataset, we sample 1024, 768, and 512 samples, and compute the $\min_{\alpha,\beta} \Delta_{l,j}^{\alpha,\beta}$, where $\alpha$ and $\beta$ are classes, $l$ denotes the layer, and $j$ denotes the filter, as defined in Section 5 of the main manuscript. We present plots for three layers below.

Table 5: Pruning ResNet Models with no training dataset and no finetuning on CIFAR-10

| Model/Param. Sparsity | Pruning Method | Acc. Drop | Comp. with ITERTVSPRUNE |
|---|---|---|---|
| ResNet18/19.12% | ITERTVSPRUNE | **1.50%** | - |
|  | Random-3 | 13.21% | -11.71% |
|  | $L_1$ | 7.88% | -6.38% |
|  | Gradient-FO | 4.56% | -3.06% |
| ResNet18/33.91% | ITERTVSPRUNE | **4.99%** | - |
|  | Random-3 | -19.89% | -15.33% |
|  | $L_1$ | 14.92% | 10.36% |
|  | Gradient-FO | -10.02% | -5.46% |
| ResNet18/39.21% | ITERTVSPRUNE | **10.0%** | - |
|  | Random-3 | 38.91% | -28.90% |
|  | $L_1$ | 28.76% | -18.76% |
|  | Gradient-FO | 18.44% | -8.43% |
| ResNet20/2.4% | ITERTVSPRUNE | **1.4%** | - |
|  | Random-3 | 5.2% | -3.8% |
|  | $L_1$ | 3.5% | -18.76% |
|  | Gradient-FO | 2.8% | -1.4% |
| ResNet20/4.8% | ITERTVSPRUNE | **4.7%** | - |
|  | Random-3 | 13.2% | -8.5% |
|  | $L_1$ | 8.9% | -4.2% |
|  | Gradient-FO | 6.23% | -1.63% |
| ResNet20/7.1% | ITERTVSPRUNE | **9.41%** | - |
|  | Random-3 | 29.2% | -19.8% |
|  | $L_1$ | 17.3% | -7.89% |
|  | Gradient-FO | 14.8 | -5.39% |
| ResNet50/24.0% | ITERTVSPRUNE | **1.50%** | - |
|  | Random-3 | 25.5% | -24.0% |
|  | $L_1$ | 15.1% | -13.60% |
|  | Gradient-FO | 9.82% | -7.32% |
| ResNet50/29% | ITERTVSPRUNE | **4.62%** | - |
|  | Random-3 | 26.2% | -21.58% |
|  | $L_1$ | 17.3% | -14.68% |
|  | Gradient-FO | 7.2% | -2.58% |
| ResNet50/34.1% | ITERTVSPRUNE | **9.94%** | - |
|  | Random-3 | 44.45% | -34.51% |
|  | $L_1$ | 24.3% | -14.36% |
|  | Gradient-FO | 14.25% | -4.31% |
| ResNet56/3.4% | ITERTVSPRUNE | **1.47%** | - |
|  | Random-3 | 7.12% | -5.65% |
|  | $L_1$ | 5.46% | -3.09% |
|  | Gradient-FO | 4.12% | -2.65% |
|  | CHIP | **1.36** | 0.11 |
| ResNet56/7.6% | ITERTVSPRUNE | **4.82%** | - |
|  | Random-3 | 23.42% | -18.60% |
|  | $L_1$ | N/A | - |
|  | Gradient-FO | 9.41% | -4.59% |
|  | CHIP | 5.56 | -0.74 |

We observe that even when we reduce the number of samples by a factor of 2, we see that the ordering of filters according to $\min_{\alpha,\beta} \Delta_{\alpha,\beta}^{l,j}$ (and thus, MinTVS$(l,j)$) is retained even for comparatively smaller sample sizes.

## E.5 EXPERIMENTS WITH FINE TUNING

In this section, we detail our results on using ITERTVSPRUNE in the setting wherein we have access to the training set and loss function, thereby enabling us to fine-tune our models after pruning. Our experiments involve fine-tuning models trained on CIFAR10 and Imagenet. Broadly speaking, our experiments demonstrate two facts.

1. Models pruned by our algorithm are able to return to the accuracy of the original pretrained model,

2. Fine-tuning our models yield models that are competitive in accuracy with those obtained using current, state-of-the-art methods such as Sui et al. (2021).

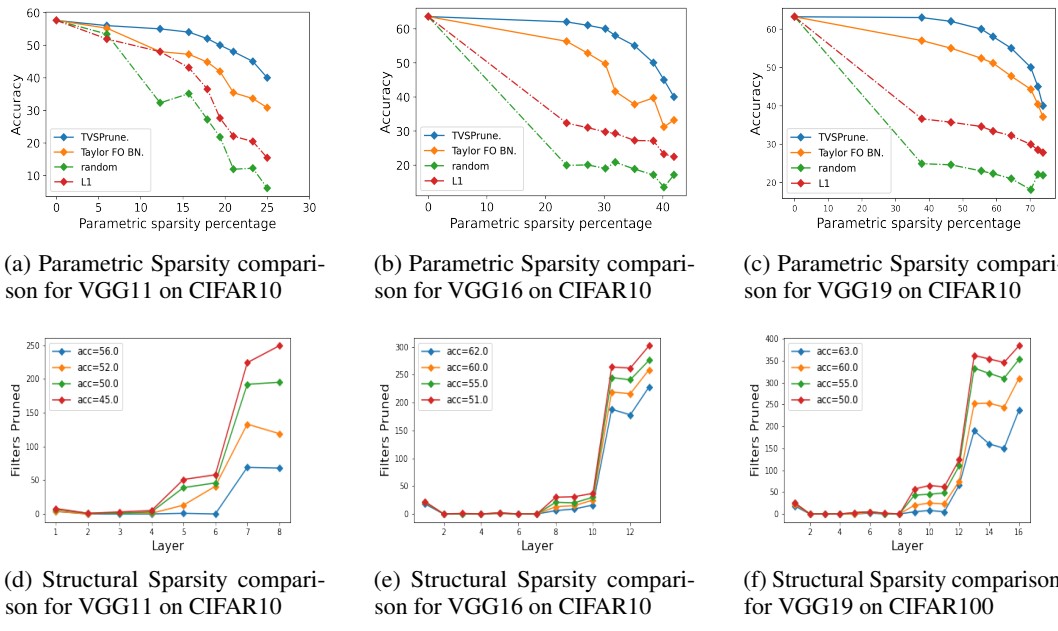

(a) Parametric Sparsity comparison for VGG11 on CIFAR10

(b) Parametric Sparsity comparison for VGG16 on CIFAR10

(c) Parametric Sparsity comparison for VGG19 on CIFAR10

(d) Structural Sparsity comparison for VGG11 on CIFAR10

(e) Structural Sparsity comparison for VGG16 on CIFAR10

(f) Structural Sparsity comparison for VGG19 on CIFAR100

Figure 8: Comparison of accuracies of different pruning algorithms applied to VGG models trained on CIFAR100

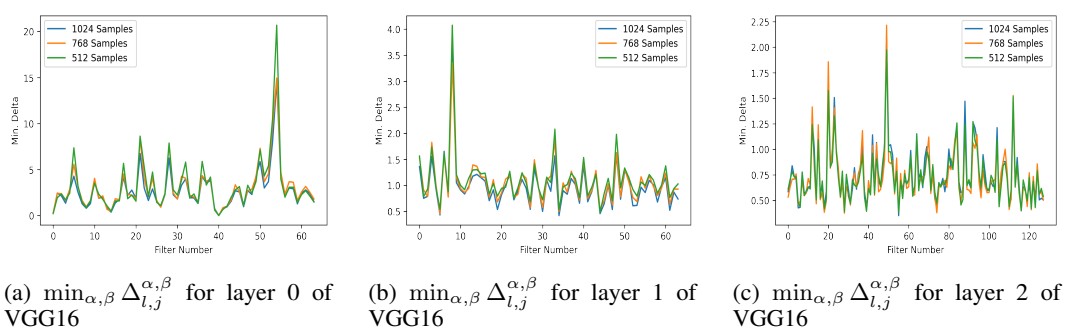

(a) $\min_{\alpha,\beta} \Delta_{l,j}^{\alpha,\beta}$ for layer 0 of VGG16

(b) $\min_{\alpha,\beta} \Delta_{l,j}^{\alpha,\beta}$ for layer 1 of VGG16

(c) $\min_{\alpha,\beta} \Delta_{l,j}^{\alpha,\beta}$ for layer 2 of VGG16

Figure 9: Effect of sample size on ranking of $\min_{\alpha,\beta} \Delta_{l,j}^{\alpha,\beta}$

### E.5.1 RESULTS OF PRUNING WITH FINE TUNING ON CIFAR10

We now present the results of fine-tuning models trained on the CIFAR10 dataset. We chose to fine-tune VGG16 models pruned using ITERTVSPRUNE and CHIP(Sui et al., 2021), and a ResNet18 model pruned with ITERTVSPRUNE. We detail our results below in Table 7.

We observe that after fine-tuning for 50 epochs, we are able to recover the accuracy of the original model, even when the model is pruned substantially, with $67.5\%$ of parameters removed, as is the case with VGG16.

### E.5.2 RESULTS OF PRUNING WITH FINE TUNING ON IMAGENET

We now present the results of fine-tuning models trained on the Imagenet dataset. We chose to fine-tune VGG16 models pruned using ITERTVSPRUNE and CHIP(Sui et al., 2021), and a ResNet18 model pruned with ITERTVSPRUNE. We detail our results below in Table 8.

As was the case with models trained on CIFAR10, we observe that our models recover the original accuracy after substantial fine-tuning. However, the extent of fine-tuning required to achieve this is substantially greater for models trained on Imagenet than for models trained on CIFAR10.

Table 6: Pruning VGG Models with no training dataset and no finetuning on CIFAR-100

| Model Name /Param. Sparsity | Pruning Method | Acc. Drop | Comp. w/ ITERTVSPRUNE |
|---|---|---|---|
| VGG-11/5.99% | ITERTVSPRUNE | **1.6 %** | - |
| | Random-3 | 4.2% | -2.6% |
| | $L_1$ | 5.7% | -4.1% |
| | Gradient-FO | 2.4% | -0.8% |
| | CHIP | - | - |
| VGG-11/17.9% | ITERTVSPRUNE | **5.6 %** | - |
| | Random-3 | 30.4% | -24.8% |
| | $L_1$ | 21.1% | -15.5% |
| | Gradient-FO | 12.8% | -7.2% |
| | CHIP | - | - |
| VGG-11/23.3% | ITERTVSPRUNE | **12.6 %** | - |
| | Random-3 | 45.4% | -32.8% |
| | $L_1$ 37.2% | -24.6% | |
| | Gradient-FO | 24.0% | -11.4% |
| | CHIP | - | - |
| VGG-16/23.6% % | ITERTVSPRUNE | **5.56 %** | - |
| | Random-3 | 42.74% | -37.18% |
| | $L_1$ | 34.26% | -28.7% |
| | Gradient-FO | 21.96% | -16.4% |
| | CHIP | - | - |
| VGG-16 / 31.9 % | ITERTVSPRUNE | **4.9%** | - |
| | Random-3 | 66.3% | -61.4% |
| | $L_1$ | 35.6% | -30.7% |
| | Gradient-FO | 31.3% | -26.4% |
| | CHIP | - | - |
| VGG-16 / 40.2% | ITERTVSPRUNE | **18.56 %** | - |
| | Random-3 | 50.08% | -31.52% |
| | $L_1$ | 40.38% | -21.82% |
| | Gradient-FO | 32.36% | -13.8% |
| | CHIP | - | - |
| VGG-19 / 37.8% | ITERTVSPRUNE | **0.2%** | - |
| | Random-3 | 38.3% | -38.1% |
| | $L_1$ | 26.6% | -26.4% |
| | Gradient-FO | 6.2% | -6.0% |
| | CHIP | - | - |
| VGG-19 59.0% | ITERTVSPRUNE | **5.2%** | - |
| | Random-3 | 40.9% | -35.7% |
| | $L_1$ | 29.8% | -24.6% |
| | Gradient-FO | 12.1% | -6.9% |
| | CHIP | - | - |
| VGG-19 72.3% | ITERTVSPRUNE | 18.2% | - |
| | Random-3 | 41.1% | -22.9% |
| | $L_1$ | 34.7% | -16.5% |
| | Gradient-FO | 22.8% | -4.6% |
| | CHIP | - | - |

Table 7: Accuracy of Fine-Tuned models trained on CIFAR10. "Acc. Drop w/o FT" refers to the accuracy drop obtained without fine-tuning the model post pruning, and "Acc. Drop w/ FT " denotes the accuracy drop after retraining the model.

| Model Name /Param. Sparsity | Pruning Method | Acc. Drop w/o FT | Acc. Drop w/ FT |
|---|---|---|---|
| VGG-16/67.5% | ITERTVSPRUNE | **12.9 %** | **0.21%** |
| VGG-16/67.5% | CHIP | 16.6% | 0.64% |
| ResNet18/39.2% | ITERTVSPRUNE | 10.0% | 0.06% |

## E.6 VALIDATING USING MinTVS FOR PRUNING

In this section, we conduct experiments validating the us MinTVS scores, as well as the key hypothesis of the paper. We show that for models trained on the CIFAR10 dataset, pruning filters with the smallest MinTVS scores achieves superior accuracy compared to pruning filters with the largest MinTVS scores. We detail this experiment below.

### E.6.1 EXPERIMENTAL SETUP

We consider VGG16 and VGG19 models trained on the CIFAR10 dataset. We consider models with LDIFF scores that are greater than 0.4; that is, $\text{LDIFF}(\eta_l, l) > 0.4$, where $\eta_l = 0.75\overline{\text{MinTVS}}(l)$ is

Table 8: Accuracy of Fine-Tuned models trained on Imagenet. "Acc. Drop w/o FT" refers to the accuracy drop obtained without fine-tuning the model post pruning, and "Acc. Drop w/ FT " denotes the accuracy drop after retraining the model.

| Model Name /Param. Sparsity | Pruning Method | Acc. Drop w/o FT | Acc. Drop w/ FT |
|---|---|---|---|
| ResNet50/24.7% | ITERTVSPRUNE | **31.2 %** | **0.03%** |
| ResNet50/24.7% | CHIP | 34.2% | 0.26% |

0.75 times the mean MinTVS score for the layer. The partition of the validation set for accuracy measurement and score computation is given in 2. We do so since the variation in MinTVS scores for layers with low LDIFF scores is minimal; thus, there is little difference between the largest scores and the smallest.

### E.6.2 EXPERIMENTAL RESULTS

We now present our experimental results. We observe that pruning discriminative filters in general yields mode dramatic drops in accuracy when compared to pruning non-discriminative filters, though the effect is slightly less for deeper layers (those closer to the output). This observation supports the hypothesis that, given layers with a mix of discriminative and non-discriminative filters, we can safely prune the non-discriminative filters, but not the former. We present these results in Figure 10.

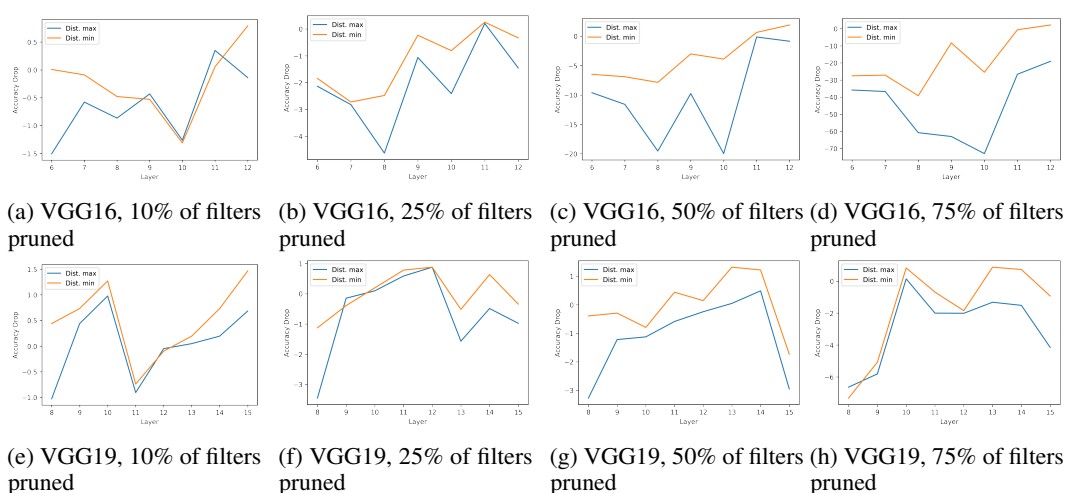

(a) VGG16, 10% of filters pruned  (b) VGG16, 25% of filters pruned  (c) VGG16, 50% of filters pruned  (d) VGG16, 75% of filters pruned

(e) VGG19, 10% of filters pruned  (f) VGG19, 25% of filters pruned  (g) VGG19, 50% of filters pruned  (h) VGG19, 75% of filters pruned

Figure 10: Comparison of accuracies of when pruning fractions of filters with least MinTVS scores with pruning fraction of filters with greatest MinTVS scores for VGG16 and VGG19 trained on CIFAR10. "Dist. max" refers to accuracies when most discriminative filters are pruned, and "Dist. min" refers to accuracies when least discriminative filters are pruned

