# OpenReview forum: "TVSPrune - Pruning Non-discriminative filters via Total Variation separability of intermediate representations without fine tuning"
_ICLR.cc/2023/Conference — ICLR 2023 poster_

### Official Review · Reviewer_tFxW · 2022-10-24

**Confidence:** 4
**Correctness:** 2
**Technical Novelty And Significance:** 2
**Empirical Novelty And Significance:** 2
**Recommendation:** 3

**Clarity, Quality, Novelty And Reproducibility:**

The writing is average, with some typos. The novelty is not significant. The reproducibility seems ok, but the experimental setting is questionable.

**Strength And Weaknesses:**

+The setting of pruning without finetuning is kind of attractive.

-The calculation of MinTVS requires samples and their labels. It's hard to understand why authors emphasize they do not need the loss function. Compared to computing loss and using related channel or filter importance scores, MinTVS may not have advantages in terms of efficiency.

-The claim of 'not using training set' is also confusing. Authors resplit the test set to calculate MinTVS. If the whole test dataset is used for evaluation, then MinTVS must use samples from the training set.

-Although the authors claim that they do not finetune the model, the performance of their method does not have significant advantages compared to regular pruning methods like CHIP.

-The experimental settings are also confusing. It's good to show accuracy without any finetuning process. However, some results are quite meaningless. For example, losing more than 30% accuracy for ResNet-50 on ImageNet when only reducing around 25% parameters. Authors should provide meaningful results by including the finetuning process.

-The proposed algorithm includes tuning several hyperparameters, and how to decide the final pruning rate is not straightforward.

**Summary Of The Paper:**

The authors proposed to solve the problem of formalizing and quantifying the discriminating ability of filters through the total variation (TV) distance between the class-conditional distributions of the filter outputs.

**Summary Of The Review:**

The setting of pruning without finetuning is interesting. But many arguments and experimental settings of this paper are not valid or confusing.

---

> ### Author Response · Authors · 2022-11-16
> **Response to Reviewer tFxW (1)**
>
> We thank the reviewer for the insight and feedback.  As suggested by the reviewer, we have conducted additional experiments to validate our results; these experiments primarily address the question of how our proposed method works in the fine-tuning/retraining regime. We hope we have addressed the concerns and questions raised by the reviewer in a satisfactory and elucidatory manner, and look forward to further engagement with the reviewer.
> 1. **The setting of pruning without finetuning is kind of attractive.**
> We thank the reviewer for this comment. First, the setting we consider is more challenging than "pruning without fine-tuning" -  we assume *that both* the training set and the loss are unavailable to us.
> Pruning without the training data or the loss function is a challenging and underexplored area of research, as noted in  [1], and has recently started to attract research attention [3,4,5]. However, they only address the problem of unstructured pruning in this setting.
>
> 2. **The calculation of MinTVS requires samples and their labels.**
> We would like to clarify that the setting we consider assumes that we possess none of the training data used to obtain the original pre-trained model. We only require samples from the same distribution, and we would like to clarify that we do not use any of the training data used to obtain the pretrained model. Rather, we only use samples from distribution used to generate the data.
> 3. **It's hard to understand why authors emphasize they do not need the loss function.**
> The goal of this work is to address the challenging problem of pruning without the training set. As noted in works such as [1,2,3], this is an important and under-researched area of inquiry. In this work, we take a step forward with the additional challenge of not having access to the loss function as well.
> 4.  **Compared to computing loss and using related channel or filter importance scores, MinTVS may not have advantages in terms of efficiency.**
> The crux of our method is the approximation of the TV distance between the class conditional distributions of the filter outputs. Aside from the computation of the minimum TV separation (MinTVS), the pruning setup is almost identical to those used in prior works, such as those listed in [1,2]. In terms of storage, our work requires only a few samples from each class in order to measure the MinTVS values for each filter. In particular, we only need to store the class conditional moments for each filter once, at runtime.
> As noted in [1], data-free pruning schemes are an important and active area of research. We motivate our work by the fact that typical data-free schemes require significant and expensive fine-tuning to recover accuracy, Our work relaxes this problem, and requires only the distributions. While we use examples from the test set in order to compute the statistical distances between the class conditional distributions of the outputs, note that we do not strictly require them - we simply require the moments of those distributions for pruning, which can be provided *a priori*.
>
> 5. **The claim of 'not using training set' is also confusing. Authors resplit the test set to calculate MinTVS. If the whole test dataset is used for evaluation, then MinTVS must use samples from the training set.**
> We would like to clarify that the training data used to obtain the original pre-trained model is not used at all in our method and that we do not use the entire test set for evaluation.  As stated previously, we only require samples from the same distribution as the training data. In this work, we partition the test set in two. We use the first partition of the test set as a proxy for samples of the data distribution, and **this partition is used exclusively for computing MinTVS scores**. The second partition is used exclusively for measuring the accuracy of the pruned models. The details of these partitions are listed in Section 7, on page 7, and in Table 2 of Section E of the Appendix, on page 19.
>
> **References**
>
> [1] Hoefler et al. *Sparsity in Deep Learning: Pruning and growth for efficient inference and training in neural networks.* 2021.
>
> [2] Blalock et al. *What is the State of Neural Network Pruning?* 2020.
>
> [3] Yin et al. *Dreaming to Distill: Data-free Knowledge Transfer via DeepInversion*. 2020.
>
> [4]  Tanaka et al. *Pruning neural networks without any data by iteratively conserving synaptic flow.* 2020
>
> [5] Gebhart et al. *A unified paths perspective for pruning at initialization*. 2021

---

> ### Author Response · Authors · 2022-11-16
> **Response to Reviewer tFxW (2)**
>
> 6. **Although the authors claim that they do not finetune the model, the performance of their method does not have significant advantages compared to regular pruning methods like CHIP.**
> Our work considers the problem of structured pruning without access to the training set or loss function. Thus, our experimental validation aims to elucidate the potential of data-free structured pruning in a similar manner to other important works in the data-free pruning space, such as Synflow [4].
> Next, our experiments aim to demonstrate that our method is at least competitive with, if not superior to, SOTA methods when the training set/loss function are unavailable. As we reported in Table 3 on page 21, our method outperforms CHIP consistently for VGG16 trained on CIFAR10, providing models with 6% better test accuracy than similar models with the same sparsity when using CHIP, when fine-tuning is not considered. We also demonstrate in Table 1 that our method is competitive with CHIP [5] in this same setting for the Imagenet dataset as well.
> 7. **The experimental settings are also confusing. It's good to show accuracy without any finetuning process. However, some results are quite meaningless. For example, losing more than 30% accuracy for ResNet-50 on ImageNet when only reducing around 25% parameters.**
> As mentioned previously, and in Sections 2 and 7.1-7.3 of the paper, the key aims of our experimental validation were to:
> a) confirm our key hypothesis about the mixture of discriminative and non-discriminative filters;
> b) show the efficacy of our proposed algorithm in the given setting, with similar experiments as proposed in [4];
> c) Show that in the regime without access to the training set and loss function, our proposed method is competitive with or superior to existing baselines. In that setting, while we observe significant accuracy losses on ResNet50 (31.4% accuracy drop with 25% sparsity), we observe that our method outperforms state-of-the-art methods such as CHIP; we showcase these results in Table 1 in Section 7.3.
>
> 8. **Authors should provide meaningful results by including the finetuning process.**
> We thank the reviewer for this suggestion. We present a brief snapshot of results with fine tuning on Imagenet and CIFAR10.
> **CIFAR10 Experiments.**
>
> | Architecture/Method | Baseline Acc.  |Acc. drop (no finetuning)  |Acc. drop (with finetuning)  | Params. removed (%)  |
> |---|---|---|---|---|
> |VGG16/TVSPrune  |  94.16| **12.9%** | **0.2%** |67.5%  |
> |  ResNet18/TVSPrune|93.07%  |  10.0% | 0.06%  | 39.2%  |
>
> **Imagenet Experiments.**
> | Architecture/Method | Baseline Acc.  |Acc. drop (no finetuning)  |Acc. drop (with finetuning)  | Params. removed (%)  |
> |---|---|---|---|---|
> |ResNet50/TVSPrune  |  76.5| **31.3%** | **0.0%** |24.75%  |
>
> We observe that even in the setting wherein we are able to fine-tune the model, we are able to recover the accuracy of the trained model. Moreover, we present a report of these results in the newly added Section E.5 of the supplemental material.
>
> 9. **The proposed algorithm includes tuning several hyperparameters**
>  IterTVSPrune has 2 hyperparameters:
> i) $\eta_0$: this is the initial threshold for discriminative ability. At iteration $k$ of Algorithm 2 (IterTVSPrune), $\eta_k= \eta_0 - k\delta_\eta$ is the threshold for discriminability.
> ii) $\delta_\eta$ is the quantity by which we reduce $\eta$ if the accuracy threshold (as described in Equation ( P ) on page 3) is violated.
> 10. **How to decide the final pruning rate is not straightforward.**
> As stated in Section 6, on page 6, the pruning rate is not set *a priori*. Rather, the pruning rate is based on $\eta_k$ and the LDIFF score, as described in Section 6. For TVSPrune, the pruning rate is implicitly set based on the choice of $\eta$, which is a threshold that we can use to determine which filters are discriminative. Additionally, the IterTVSPrune iteratively lowers $\eta$ (thereby increasing the fraction of discriminative filters) while checking the accuracy constraint as described in Equation P, thus increasing the total extent that filters are pruned. This is described in detail in Section 6 of the main document.
> Thus, in Algorithms 1 (TVSPrune) and 2 (IterTVSPrune), we do not explicitly provide the pruning rate; rather the number of filters pruned is based on the numbers of discriminative and non-discriminative filters for the threshold $\eta$ and an accuracy constraint as defined in Equation P.
>
>
> **References**
>
> [1] Hoefler et al. *Sparsity in Deep Learning: Pruning and growth for efficient inference and training in neural networks.* 2021.
>
> [2] Blalock et al. *What is the State of Neural Network Pruning?* 2020.
>
> [3] Yin et al. *Dreaming to Distill: Data-free Knowledge Transfer via DeepInversion*. 2020.
>
> [4]  Tanaka et al. *Pruning neural networks without any data by iteratively conserving synaptic flow.* 2020
>
> [5] Sui et al. *CHIP: CHannel Independence-based Pruning for Compact Neural Networks.* 2021

---

> > ### Comment · Reviewer_tFxW · 2022-12-07
> > **Thanks for the responses**
> >
> > I want to thank the authors for their responses. However, I still have some concerns.
> >
> > The setting of this paper did not use any training samples, but you need samples from the same data distribution. From Table.2 in the appendix, you need 7500~40000 samples for different datasets. Compared to directly using training samples, does this setting really more practical?
> >
> > Structural pruning without training data is actually studied in "Dreaming to Distill" [1]. In section 4.3 and Table.5 of [1], they show the results of structural pruning without using any data. Instead, they use synthetic samples generated from their method. Their pruning method is based on [2], and it achieves 74.0% Top-1 accuracy by pruning more than 25% FLOPs. The whole process, according to their paper, does not require the training dataset.
> >
> > In the setting of this paper, if removing access to the training dataset, IterTVSPrune prunes 25% of parameters with 31.3% Top-1 accuracy lost, which is much lower than the results reported in [1].
> >
> > [1] Yin, Hongxu, et al. "Dreaming to distill: Data-free knowledge transfer via deepinversion." Proceedings of the IEEE/CVF Conference on Computer Vision and Pattern Recognition. 2020.
> >
> > [2] Molchanov, Pavlo, et al. "Importance estimation for neural network pruning." Proceedings of the IEEE/CVF Conference on Computer Vision and Pattern Recognition. 2019.

---

> > > ### Author Response · Authors · 2022-12-08
> > > **Response to additional questions from Reviewer tFxW**
> > >
> > > We thank the reviewer for the additional feedback and discussion. We sincerely hope that our responses have addressed the reviewer's concerns, and are eager for any further discussion.
> > >
> > > 1. **The setting of this paper did not use any training samples, but you need samples from the same data distribution. From Table.2 in the appendix, you need 7500~40000 samples for different datasets. Compared to directly using training samples, does this setting really more practical?**
> > > As noted in [1,3,4], Data-Free pruning is an important and active area of research. In this work, we relax the problem to that of *distributional pruning*, wherein we have access to the moments of the class-conditional distributions, but not access to the training set or loss function. To obtain rough estimates of the class-conditionals, we require 40 samples per class (which we established empirically); this leads to 40000 samples for Imagenet, but far fewer for CIFAR10 and CIFAR100. This is a relatively small number of samples in high dimensional spaces (such as images), and thus, is practical in the training-free regime.
> > >
> > >
> > > 2. **Structural pruning without training data is actually studied in "Dreaming to Distill" [1]. In section 4.3 and Table.5 of [1], they show the results of structural pruning without using any data. Instead, they use synthetic samples generated from their method.**
> > > In our proposed method, we do not require synthetic samples at all; rather, we simply require the moments of the class conditional distributions of the filter outputs. This obviates the cost of generating synthetic samples, which can be extremely expensive ([1] requires 2800 hours on a V100 GPU to generate sufficiently many samples).
> > > Moreover, our proposed methodology is grounded in the distributional information of the filter outputs, and can be adopted toward pruning under a variety of other constraints, including transfer learning and class addition; these are areas of future work.
> > >
> > > 3. **Their pruning method is based on [2], and it achieves 74.0% Top-1 accuracy by pruning more than 25% FLOPs.**
> > > The pruning method proposed in [1] relies on the synthetic samples generated to inform the pruning strategy. Using the synthetic samples as a surrogate training set along with the loss function, the gradient-based pruning strategy proposed in[2] is used to prune, and then fine-tune the model. This differs from our work, as one of the key assumptions we make is that we do not have access to either the training set or the loss function.
> > > 4. **The whole process, according to their paper, does not require the training dataset. In the setting of this paper, if removing access to the training dataset, IterTVSPrune prunes 25% of parameters with 31.3% Top-1 accuracy lost, which is much lower than the results reported in [1].**
> > > In our previous response on Nov. 16, we presented results in the regime wherein we have access to the training data and the loss function. We observed that our method recovered the full accuracy of the original, unpruned model when 25% of the parameters were pruned, which is superior to the results given in [1]. As mentioned previously, these results were not presented in the main paper, as they are contrary to the key setting of our investigation - that is, where access to both the training set and the loss function is unavailable.
> > >
> > >
> > >
> > > [1] Yin, Hongxu, et al. "Dreaming to distill: Data-free knowledge transfer via deepinversion." Proceedings of the IEEE/CVF Conference on Computer Vision and Pattern Recognition. 2020.
> > >
> > > [2] Molchanov, Pavlo, et al. "Importance estimation for neural network pruning." Proceedings of the IEEE/CVF Conference on Computer Vision and Pattern Recognition. 2019.
> > >
> > > [3] Tanaka et al. *Pruning neural networks without any data by iteratively conserving synaptic flow.* 2020
> > >
> > > [4] Sui et al. *CHIP: CHannel Independence-based Pruning for Compact Neural Networks.* 2021

---

### Official Review · Reviewer_Ghni · 2022-10-25

**Confidence:** 4
**Correctness:** 3
**Technical Novelty And Significance:** 3
**Empirical Novelty And Significance:** 3
**Recommendation:** 8

**Clarity, Quality, Novelty And Reproducibility:**

Clarity
The context, problem domain and intent of the paper is well put forward and easy to digest.

Quality and Originality
While the finer points of the technical contribution are novel, the broader context and its viability are concerning, given the large amount of existing work in pruning and the current state of the domain.



**Strength And Weaknesses:**

Strengths
- The technical contribution and its relevant information have been explained well.
- The overall structure of the manuscript is solid and helps the reader digest the information in a steady manner.

Weaknesses
- Could the authors comment on how to connect the proposed relaxation back to data-free structured pruning? (Pg. 2, Paragraph 1, Line 2-5)
- Could the authors contextualize the theoretical setting assumed in the proposed work, of class conditioned distribution from various layers of the network, given that storing such information consumes extra memory? In addition, when iteratively updating the pruning algorithm, how much do the distributions vary, post pruning of the previous layer? Are their effects exaggerated on the distribution of successive layers?
- Could the authors provide more insight in choosing Random Pruning as a baseline to understand the impact of TV distance? Since due to its weak assumptions it provides loose bounds and a measure that provides tighter bounds could inform the pruning better.
- Could the authors clarify and define the meaning of the notations used in Equation P?
- Could the authors describe and discuss in detail the statement "Note that we also observe that some layers have features that cannot discriminate well, and yet cannot be pruned."?
- Could the authors explain how values of $\eta$ were obtained?
- From an experimental perspective, could the authors clarify whether selecting the filter with the highest  TV distance matched with the largest drop in performance? (i.e., the values of TV distance corresponded to the expected ranking of discriminative filters)
- Could the authors clarify if a batch size of 30000 was used for experiments in Section 7.2?
- Given the extreme drop in performance, relative to a small amount of parameters being removed (Table 1), could the authors discuss if TV distance and LDIFF could be further highlighted when framed similar to more standard pruning setups?

Post Rebuttal
- Based on the responses provided by the authors, I have updated the final recommendation.

**Summary Of The Paper:**

The proposed work uses the hypothesis: Non-discriminative filters do not contribute a lot to the predictive performance of a network and can be removed safely. Keeping this in mind, along with a privacy based setting where the original training data and loss function are unavailable yet the a sample from the original distribution is available, the proposed work uses TotalVariation distance between class conditioned distributions as a measure to identify discriminative filters. In addition, the proposed work offers LDIFF score as a way to ascertain the extent to which a layer possesses a mixture of discriminative and non-discriminative filters. Overall, these two ideas are combined to provide a pruning algorithm.

**Summary Of The Review:**

Certain choices in parameters and rationalization behind the problem domain are the main concerns when it comes to the proposed work. Addressing the weaknesses highlighted above should solves these concerns.

---

> ### Author Response · Authors · 2022-11-16
> **Response to Reviewer Ghni (1)**
>
> We thank the reviewer for the insightful comments and questions. In order to respond to the reviewer's concerns, we have conducted a number of additional experiments, which we describe below, and are presented with greater detail in sections E.6 and E.7 of the supplementary material, which we shall update tomorrow.
>
>
> 1.   **Could the authors comment on how to connect the proposed relaxation back to data-free structured pruning? (Pg. 2, Paragraph 1, Line 2-5)**
> Data-free structured pruning reflects involves pruning without the training data. We advance this problem by assuming that we do not have access to the loss function either. We then relax the problem by assuming that, despite not having access to the loss function or training set, we have access to the distribution from which the training set was drawn, e.g. pictures of cats and dogs.
> 2.  **Could the authors contextualize the theoretical setting assumed in the proposed work, of class conditioned distribution from various layers of the network, given that storing such information consumes extra memory?**
> The key goal of this work is to address the problem of pruning neural networks without access to the training set or loss function. We relax this problem, allowing ourselves access to the distributions of the data distribution; we call this paradigm *distributional pruning*.
> As mentioned in Sections 1,4,5 and 6 of the paper, in order to measure the TV distance between the class conditional distributions, we need to store the moments of the class conditional distributions.  This does not require any additional storage, except during runtime for the computation of the MinTVS scores, where distributional moments are stored for only a single pass of the algorithm and then discarded.
> However, our goal is to measure the *relative* ranking of filters using the minimum TV separation; we use the Hellinger lower bound for this, enabling us to use the moments of the class conditionals, under the Gaussianity assumption.
> 3. **In addition, when iteratively updating the pruning algorithm, how much do the distributions vary, post pruning of the previous layer? Are their effects exaggerated on the distribution of successive layers?**
> Our algorithm prunes each layer sequentially in order to capture the effect of pruning previous layers on the "current" layer, as we hypothesized that pruning earlier layers could impact the class-conditional distributions of filter outputs of the subsequent layers. Thus, in Algorithm 1 (TVSPrune), we only prune layer $l$ after pruning layers $1,\cdots, l-1$. However, the key fact to note is that **we are chiefly concerned with the TV distance between the distributions, and not the distributions themselves.** Thus, even if the distributions change, the *ranking* of filters in terms of the MinTVS values are still relatively stable.
>
> We hope that our response has sufficiently assuaged the reviewer's concerns. We once again thank the reviewer for the thorough reading and insightful comments, and look forward to additional engagement with the reviewer.
>
> **References**
>
> [1] Hoefler et al. *Sparsity in Deep Learning: Pruning and growth for efficient inference and training in neural networks.* 2021.
>
> [2] Blalock et al. *What is the State of Neural Network Pruning?* 2020.
>
> [3] Tanaka et al. *Pruning neural networks without any data by iteratively conserving synaptic flow.* 2020
>
> [4] Gebhart et al. *A unified paths perspective for pruning at initialization*. 2021
>
> [5] Sui et al. *CHIP: CHannel Independence-based Pruning for Compact Neural Networks.* 2021
>
> [6] Li et al. *Pruning Filters for Efficient ConvNets.* 2016
>
> [7] Yin et al. *Dreaming to Distill: Data-free Knowledge Transfer via DeepInversion*. 2020.
>
> [8] Shen et al. *Structural Pruning via Latency-Saliency Knapsack*. 2022.
>
> [9] Anderson et al. *The High-Dimensional Geometry of Binary Neural Networks*. 2018.

---

> > ### Comment · Reviewer_Ghni · 2022-11-19
> > **Thank you for the responses**
> >
> > I would like to thank the authors for their detailed feedback to the original comments.
> > Based on the feedback, I had a couple of clarifications,
> > - The storage of moments, even if it only during runtime, is still a cost overhead that needs to be kept in mind when choosing/executing the algorithm presented in the manuscript (in terms of overhead/complexity needed).
> > - When discussing the random pruning, my intention in saying "loose bounds" was to show that while random pruning offers an unguided approach to indicating the effectiveness of pruning, a more directed approach could provide tight bounds on just how much different layers can/cannot be pruned. There could possibly be a correlation in the choice of pruning constraint and internal scores computed in the algorithm.
> > - The initial layers seem to have an odd level of behavior, w.r.t. the proposed algorithm. Could the authors further clarify the statement " For layers with low LDIFF scores (such as the initial layers), we observe little difference between pruning the most discriminative and least discriminative filters. ". This statement seems to indicate that regardless of which filters are pruned, the network is able to sustain performance. This is counter to the intuition that initial layers grasp basic concepts which form the necessary building blocks for subsequent layers.

---

> > > ### Author Response · Authors · 2022-11-19
> > > **Response to Reviewer Ghni**
> > >
> > > We thank the reviewer for the additional discussion, and we offer the following clarifications. We hope we have answered the reviewer's questions in a clear and satisfactory manner, and are eager for further discussion.
> > >
> > > 1.   **The storage of moments, even if it only during runtime, is still a cost overhead that needs to be kept in mind when choosing/executing the algorithm presented in the manuscript (in terms of overhead/complexity needed).**
> > > We note that the runtime overhead is independent of the number of layers, as we execute the algorithm sequentially.
> > > To see this, suppose we have a model with a total of $N$ filters in the network. Then,  at runtime, we need only store at most $fN$ filter outputs per layer, where $f$ is a fraction of the total number of filters. For VGG16, for instance, this value is 0.12. This is similar to the storage requirements in works such as [8].
> > > 2.  **When discussing the random pruning, my intention in saying "loose bounds" was to show that while random pruning offers an unguided approach to indicating the effectiveness of pruning, a more directed approach could provide tight bounds on just how much different layers can/cannot be pruned.**
> > > As stated in our previous response, we chose to use random pruning in our experiments simply because it offers a worst-case baseline, as noted in a variety of works such as [1,2,3].
> > > 3. **There could possibly be a correlation in the choice of pruning constraint and internal scores computed in the algorithm.**
> > > We thank the reviewer for this insightful comment. However, random pruning is still being actively investigated, and yielding useful insights [5,6,7]. Thus,  this question requires much deeper investigation, and thus, we feel that it is outside the scope of our present work.
> > > 4.  **The initial layers seem to have an odd level of behavior, w.r.t. the proposed algorithm. Could the authors further clarify the statement " For layers with low LDIFF scores (such as the initial layers), we observe little difference between pruning the most discriminative and least discriminative filters. ". This statement seems to indicate that regardless of which filters are pruned, the network is able to sustain performance. This is counter to the intuition that initial layers grasp basic concepts which form the necessary building blocks for subsequent layers.**
> > >  We clarify that our statement is meant to indicate that irrespective of which filters are pruned, the network is *unable* to sustain performance. A possible rephrasing of the quoted sentence, for better clarity, would be:
> > > "For layers with low LDIFF scores (such as the initial layers), pruning the most discriminative and least discriminative filters both lead to severe losses in performance."
> > > Additionally, we note that the LDIFF score is meant to indicate whether a layer is easy to prune (high score) or difficult to prune (low score), by capturing the mix of discriminative and non-discriminative filters. Since initial layers for a variety of datasets and architectures have low LDIFF scores, (thus indicating that they are hard to prune), this is in concurrence with the intuition that initial layers grasp crucial features, and are thus hard to prune. This is also supported by our experiments in Section 7.2.
> > >
> > > [1] Hoefler et al. *Sparsity in Deep Learning: Pruning and growth for efficient inference and training in neural networks.* 2021.
> > > [2] Tanaka et al. *Pruning neural networks without any data by iteratively conserving synaptic flow.* 2020
> > > [3] Li et al. *Pruning Filters for Efficient ConvNets.* 2016
> > > [4] Gebhart et al. *A unified paths perspective for pruning at initialization*. 2021
> > > [5] Li et al . *Revisiting Random Channel Pruning for Neural Network Compression*. 2022
> > > [6] Liu et al. *The unreasonable effectiveness of random pruning: Return of the most naive baseline for sparse training*. 2022.
> > > [7] Su et al. *Sanity-checking pruning methods: Random tickets can win the jackpot* 2020.
> > > [8] Sui et al. *CHIP: CHannel Independence-based Pruning for Compact Neural Networks.* 2021

---

> ### Author Response · Authors · 2022-11-16
> **Response to Reviewer Ghni (2)**
>
>
> 4.   **Could the authors provide more insight in choosing Random Pruning as a baseline to understand the impact of TV distance? Since due to its weak assumptions it provides loose bounds and a measure that provides tighter bounds could inform the pruning better.**
> We choose random pruning, wherein filters are chosen to be pruned uniformly at random, as a baseline since, as is the case in works such as [3,4,6] and as noted in [1], it offers a minimal baseline against which the performance of a pruning algorithm can be measured against. We implement this algorithm directly with the implementation given by PyTorch, without any modifications. Random pruning does not inform the TVSPrune and IterTVSPrune algorithms in any way at all. Also, we are unclear about what the reviewer is referring to with the term "loose bounds" - could the reviewer elaborate further?
> Next, as we note in Section 7.2, we use random pruning to help empirically indicate the extent to which layers can be pruned as well. If, on average, a layer can be randomly pruned without a dramatic impact on the model accuracy, it indicates that a significant fraction of non-discriminative filters is present in that layer. Conversely, if, on average, random pruning of a fixed fraction of filters results in a significant drop in accuracy, it indicates that the number of non-discriminative filters is relatively small, thereby indicating that the layer cannot be extensively pruned.
> 5.   **Could the authors clarify and define the meaning of the notations used in Equation P?**
> Equation P formalizes the specific pruning problem we address in this work. Informally, our aim is to "find the most sparse model that satisfies an accuracy constraint." Here $\mathcal{W}$ are the model parameters, and $\|\mathcal{W}\|_0$ uses the $\ell_0$ norm to denote sparsity. $f(\mathcal{W})$ denotes the test error, and $t$ is the largest acceptable test error (i.e. 5% test error, corresponding to 95% accuracy).
> 6.   **Could the authors describe and discuss in detail the statement "Note that we also observe that some layers have features that cannot discriminate well, and yet cannot be pruned."?**
> It has been well-studied that the initial layers of convolutional networks are important for generating useful representations that can be used by subsequent layers, as detailed in [8,9]. As further noted in [9, section 4], these layers should not be compressed in order to maintain the model's accuracy.
> This, therefore, raises the interesting challenge of using our distributional pruning framework to identify such layers. In the course of our experiments, we observe that there are some layers that do not contain a mix of discriminative and non-discriminative filters - that is, the MinTVS values are both very low and very similar; typically, these are the initial layers in models.  This leads us to devise the LDIFF score, as detailed in section 5 of the manuscript.
>
> 7.   **Could the authors explain how values of  η  were obtained?**
> The values of $\eta$ are generally chosen empirically. As mentioned in Section 4, for VGG models trained on CIFAR10, choosing $\eta = .05$ was effective. However, as mentioned previously, this value will change from dataset to dataset, and architecture to architecture. As we discuss in Section 5, and in Section C of the appendix, an alternate means of selecting $\eta$ is to set a different value of $\eta$ for each layer, for instance, by choosing some fraction of the mean.
>
> **References**
>
> [1] Hoefler et al. *Sparsity in Deep Learning: Pruning and growth for efficient inference and training in neural networks.* 2021.
>
> [2] Blalock et al. *What is the State of Neural Network Pruning?* 2020.
>
> [3] Tanaka et al. *Pruning neural networks without any data by iteratively conserving synaptic flow.* 2020
>
> [4] Gebhart et al. *A unified paths perspective for pruning at initialization*. 2021
>
> [5] Sui et al. *CHIP: CHannel Independence-based Pruning for Compact Neural Networks.* 2021
>
> [6] Li et al. *Pruning Filters for Efficient ConvNets.* 2016
>
> [7] Yin et al. *Dreaming to Distill: Data-free Knowledge Transfer via DeepInversion*. 2020.
>
> [8] Shen et al. *Structural Pruning via Latency-Saliency Knapsack*. 2022.
>
> [9] Anderson et al. *The High-Dimensional Geometry of Binary Neural Networks*. 2018.

---

> ### Author Response · Authors · 2022-11-16
> **Response to Reviewer Ghni (3)**
>
> 8.   **From an experimental perspective, could the authors clarify whether selecting the filter with the highest TV distance matched with the largest drop in performance? (i.e., the values of TV distance corresponded to the expected ranking of discriminative filters)**
> We thank the reviewer for another very insightful question. We present a series of new experiments investigating this question.  We present these experiments for VGG16 and VGG19 models trained on the CIFAR10 dataset in the newly added Section E.6 of the Supplementary Material.
> In our experiments, we make the following observations.
> a. For layers with low LDIFF scores (such as the initial layers), we observe little difference between pruning the most discriminative and least discriminative filters.
> b. For layers with higher LDIFF scores, we observe that pruning the most discriminative filters yields a greater loss of accuracy than pruning the least discriminative filters.
> These experiments support the hypothesis stated in the manuscript - that layers with a mixture of discriminative and non-discriminative filters can be pruned effectively by removing non-discriminative filters.
>
> 9.   **Could the authors clarify if a batch size of 30000 was used for experiments in Section 7.2?**
> We apologize for the typographical error. In section 7.2, we use the CIFAR10 and CIFAR100 datasets. For CIFAR10, we use 512 samples to estimate the MinTVS- and LDIFF scores. For CIFAR100, we used 4096 samples (to ensure we have, on average, 40 samples per class) to measure the MinTVS- and LDIFF scores. We have amended the manuscript to reflect these corrections.
>
>
> 10.  **Given the extreme drop in performance, relative to a small amount of parameters being removed (Table 1), could the authors discuss if TV distance and LDIFF could be further highlighted when framed similar to more standard pruning setups?**
> The TV-Distance and LDIFF score can be used in conjunction with any standard pruning setup. First, our use of the class conditional distributions can be used to derive saliency measures. For example, rather than using TV distance, other divergences can be used to identify important filters.
> Second, the LDIFF score can be used to inform pruning ratios. Most state-of-the-art layerwise pruning techniques rely on the user providing a pruning ratio.  As we detail in Section 7.2, LDIFF scores correlate well with the extent to which a layer can be pruned. Thus, irrespective of the method used to rank filters, LDIFF scores can be used to inform the pruning rate.
>
>
> **References**
>
> [1] Hoefler et al. *Sparsity in Deep Learning: Pruning and growth for efficient inference and training in neural networks.* 2021.
>
> [2] Blalock et al. *What is the State of Neural Network Pruning?* 2020.
>
> [3] Tanaka et al. *Pruning neural networks without any data by iteratively conserving synaptic flow.* 2020
>
> [4] Gebhart et al. *A unified paths perspective for pruning at initialization*. 2021
>
> [5] Sui et al. *CHIP: CHannel Independence-based Pruning for Compact Neural Networks.* 2021
>
> [6] Li et al. *Pruning Filters for Efficient ConvNets.* 2016
>
> [7] Yin et al. *Dreaming to Distill: Data-free Knowledge Transfer via DeepInversion*. 2020.
>
> [8] Shen et al. *Structural Pruning via Latency-Saliency Knapsack*. 2022.
>
> [9] Anderson et al. *The High-Dimensional Geometry of Binary Neural Networks*. 2018.

---

### Official Review · Reviewer_gKo4 · 2022-10-25

**Confidence:** 3
**Correctness:** 3
**Technical Novelty And Significance:** 2
**Empirical Novelty And Significance:** 3
**Recommendation:** 6

**Clarity, Quality, Novelty And Reproducibility:**

The manuscript is easy to read and is of high quality but requires more experimental results and explanations. The proposed new pruning paradigm called distributional pruning is novel. The idea of data-free pruning is not very novel. The manuscript provides enough experimental details to reproduce.

**Strength And Weaknesses:**

## Strength:

1. The authors propose a new paradigm called distributional pruning for pruning neural networks. It measures the discriminating ability of filters through the total variation (TV) distance between the class-conditional distributions of the feature maps outputs by the filters.

2. Experiments on the image classification task demonstrate the effectiveness of the proposed method.

3. The manuscript is easy to read and provides enough experimental details to reproduce.

## Weakness:

1. The proposed method may be hard to be applied in real data-free scenarios because it needs to assess thousands of images from the test set (3/4 of the training set) to calculate the distributions of feature maps. Can the proposed method use less number of images (or images from datasets with similar distribution) to calculate the distributions of feature maps? More explanations and experiments are required.

2. In Algorithm 1, the authors set the weights of the pruned filters as zero rather than remove them directly. In this way, the pruned model may still cost the same storage space to store these zero weights and consume the same computation cost to forward with the zero weights. How does the pruned model reduce the storage space and the inference time? More explanations and experiments are required.

3. In Section 5, the motivation of proposing LDIFF metric to identify which layers can be pruned is unclear. The authors claim that LDIFF metric tends to prune layers with a mixture of discriminative and non-discriminative filters and avoid pruning layers with a majority of discriminative filters or non-discriminative filters. Why should the LDIFF metric avoid pruning layers with a majority of non-discriminative filters? It would be better to visualize and analyze the fraction of discriminative filters $\tao(\eta)$ in each layer.

4. In Definition 1, the meaning of the function "sup|·|" in the equation is confusing. It would be better for the authors to explain the meaning of "sup|·|".

5. In Section 7, it would be better for the authors to provide the theoretical complexity measure and the time consumption of the proposed method.

6. The authors only show the results on heavy-weight models such as ResNet. It would be better for the authors to conduct more experiments on light-weight models such as MobileNet-V2 [1].

7. More ablations of the $\eta$ are required because the influence of $\eta$ is unclear. How does this hyper-parameter affect the performance of the proposed pruning method?

8. The authors only present the experimental results on image classification task. It would be better for the authors to show the experimental results on more computer vision tasks such as object detection and semantic segmentation.

## Minor Issues：
1. Many references in this paper have been officially published, such as “Baykal et al. (2018)” and “Frankle & Carbin (2018)”. Please reference these papers in the formal format.

2. There are some typos in this paper:
(1) In Section 7.1, "in order to reduce the test error" should be "in order to increase the test error"?
(2) In Section 6.1, "Therefore, we use the LDIFF scores to decide We now present the TVSPRUNE algorithm." is an incomplete sentence.

## Reference:
[1] MobileNetV2: Inverted Residuals and Linear Bottlenecks. CVPR 2018.


**Summary Of The Paper:**

The authors propose a training-data free structure pruning method to prune deep neural networks. Specifically, the authors propose to measure the discriminating ability of filters through the total variation (TV) distance between the class-conditional distributions of the feature maps outputs by the filters. Then, based on the above TV-distance, the authors define the LDIFF score to decide the pruning ratio of each layer. Last, the authors proposed IterTVSPrune, which iteratively prunes the model to achieve greater sparsity. Experimental results on CIFAR10/100 and ImageNet demonstrate the effectiveness of the proposed method. However, there are still some issues in the paper. Detailed comments are as follows.

**Summary Of The Review:**

The authors propose a training-data free structure pruning method to prune deep neural networks. The proposed new paradigm called distributional pruning for pruning neural networks is novel. The idea of data-free pruning is not very novel. The manuscript is easy to read and provides enough experimental details to reproduce. Experiments on the image classification task demonstrate the effectiveness of the proposed method. However, the manuscript still needs more experimental results and explanations.

---

> ### Author Response · Authors · 2022-11-16
> **Response to Reviewer gKo4 (1)**
>
> We thank the reviewer for the insightful questions, and we hope to clarify the questions raised with our response below. Based on the reviewer's feedback,  we have conducted additional experiments and made appropriate additions to the supplementary material (which we shall update tomorrow), and we have planned additional experiments as well.
>
>
> 1.  **The proposed method may be hard to be applied in real data-free scenarios because it needs to assess thousands of images from the test set (3/4 of the training set) to calculate the distributions of feature maps.**
> Our aim in this work is to measure the TV separation between class-conditional outputs of filters, which we accomplish by measuring moments and using the Hellinger lower bound, while using samples from the data distribution.  However, any samples from the data distribution are sufficient.  Since obtaining such samples is highly impractical, we elected to use a subset of the test set as a proxy for samples from a similar distribution.
> We recall that the key hypothesis of this work, stated in Hypothesis 1, is that layers that can be effectively pruned contain a mix of discriminative and non-discriminative filters, of which, the non-discriminative filters can be removed without significant loss of accuracy. Thus, to identify discriminative filters, we require only to measure the separation between the class-conditional distributions.
> 2. **Can the proposed method use less number of images (or images from datasets with similar distribution) to calculate the distributions of feature maps? More explanations and experiments are required.**
> We observe that MinTVS scores do not vary significantly when the number of samples is reduced from 1024 (approx 100 samples per class) to 512 (approx 50 samples per class), as we demonstrate in Figure 9 Section E.5 of the Supplementary material.  Our goal is to measure the relative ranking of filterwise MinTVS scores, for which a huge number of samples may not be necessary.  Thus, we observe that it is possible to prune effectively using the TVSPrune algorithm even with limited samples for each class.
>
> 3.  **In Algorithm 1, the authors set the weights of the pruned filters as zero rather than remove them directly. In this way, the pruned model may still cost the same storage space to store these zero weights and consume the same computation cost to forward with the zero weights. How does the pruned model reduce the storage space and the inference time? More explanations and experiments are required.**
> For exposition purposes, we follow the standard used in a variety of state-of-the-art works, including, [1,2,3], [4, Tables 1 and 2], [5, Table 1], wherein parametric sparsity is a commonly used surrogate for memory footprint and inference time. Thus, as we simply set the weights and biases to zero, we can:
> a. roughly infer the performance and memory footprint improvements by counting the FLOPS and parameters in the model.
> b. obtain low-memory and fast models by initializing new architectures, wherein each layer contains only the unpruned filters of the original. This new network is identical to the model with weights set to zero.
> However, we are presently conducting experiments to verify the actual inference times of some of our pruned models, and will report them shortly.
>
> 4. **In Section 5, the motivation of proposing LDIFF metric to identify which layers can be pruned is unclear.**
> As has been observed previously [7], and confirmed by in our experiments, not all layers can be pruned to the same degree while achieving minimal losses in accuracy. The goal of the LDIFF score is to identify those layers that *cannot* be pruned effectively.
>
>
> [1] Hoefler et al. *Sparsity in Deep Learning: Pruning and growth for efficient inference and training in neural networks.* 2021.
>
> [2] Blalock et al. *What is the State of Neural Network Pruning?* 2020.
>
> [3] Tanaka et al. *Pruning neural networks without any data by iteratively conserving synaptic flow.* 2020
>
> [4] Sui et al. *CHIP: CHannel Independence-based Pruning for Compact Neural Networks.* 2021
>
> [5] Li et al. *Pruning Filters for Efficient ConvNets.* 2016
>
>
> [6] Tsybakov. *Introduction to Nonparametric Estimation*. 2003.
>
> [7] Shen et al. *Structural Pruning via Latency-Saliency Knapsack*. 2022.

---

> ### Author Response · Authors · 2022-11-16
> **Response to Reviewer gKo4 (2)**
>
> 5. **The authors claim that LDIFF metric tends to prune layers with a mixture of discriminative and non-discriminative filters and avoid pruning layers with a majority of discriminative filters or non-discriminative filters. Why should the LDIFF metric avoid pruning layers with a majority of non-discriminative filters? It would be better to visualize and analyze the fraction of discriminative filters  \tao(η)  in each layer.**
> The reason for requiring a mix of discriminative and non-discriminative filters is the fact that given a threshold of discriminative ability, if there does exist a mix of discriminative and non-discriminative filters, the non-discriminative filters can be pruned without suffering significant losses in accuracy, as those filters do not contribute significantly to the classification accuracy of the model.
> On the other hand, if a layer possesses a majority of discriminative filters pruning even a few filters can result in a dramatic loss of accuracy. We observe this in our experiments with a ResNet20 model trained on CIFAR10, wherein removing even a few filters in any layer results in a dramatic drop in performance, provided no fine-tuning or retraining takes place. We detail these experiments in Table 4 for the Supplementary Material, on page 23.
> Last, assuming we have a valid threshold of discriminative ability, if a filter contains *only* non-discriminative filters,  it indicates that those filters cannot be pruned as well, as they may serve in generating useful features for subsequent layers.
>
> 6.  **In Definition 1, the meaning of the function "sup|·|" in the equation is confusing. It would be better for the authors to explain the meaning of "sup|·|".**
>     Definition 1 contains the standard, formal definition of the Total Variation Distance between two probability measures. For two measures P(.) and Q(.), |P(A)-Q(A)| defines the difference in probability assigned to an event A by P and Q. The "sup |P(A)-Q(A)|" captures the largest such distance over all events A. We refer the reviewer to [6] for further details and exposition.
>
> 7.  **In Section 7, it would be better for the authors to provide the theoretical complexity measure and the time consumption of the proposed method.**
> First, we would like to clarify that TVSPrune and IterTVS prune are identical in terms of complexity as most contemporary pruning algorithms. The only difference lies in the computation of the MinTVS scores.  For a pair of classes, computing the scores is straightforward, and requires simple arithmetic operations. The storage requirements for a model with $L$ layers trained on a dataset with $k$ classes is
> 	$Memory = k\Sigma_{l=1}^{L}n_l(1+m_l)$, where $n_l$ is the number of filters per class, and $m_l$ is the number of parameters in each filter in each layer. This is because, for each filter with $m_l$ parameters, we need to store $k$ class-conditional means (of size $m_l$), and $k$ variances, that are scalar.	The only complexity involved in computing the MinTVS score is the complexity of the sorting method.
> 	Next, it is straightforward to see that our proposed algorithm, IterTVSPrune (Algorithm 2, presented on page 7 and described in Section 6 on page 6), requires at least $(\eta_0 - \eta_{min}) / \delta_\eta$ iterations of TVSPrune (Algorithm 1, presented on page 7).
>
> **References**
>
> [1] Hoefler et al. *Sparsity in Deep Learning: Pruning and growth for efficient inference and training in neural networks.* 2021.
>
> [2] Blalock et al. *What is the State of Neural Network Pruning?* 2020.
>
> [3] Tanaka et al. *Pruning neural networks without any data by iteratively conserving synaptic flow.* 2020
>
> [4] Sui et al. *CHIP: CHannel Independence-based Pruning for Compact Neural Networks.* 2021
>
> [5] Li et al. *Pruning Filters for Efficient ConvNets.* 2016
>
>
> [6] Tsybakov. *Introduction to Nonparametric Estimation*. 2003.
>
> [7] Shen et al. *Structural Pruning via Latency-Saliency Knapsack*. 2022.

---

> ### Author Response · Authors · 2022-11-16
> **Response to Reviewer gKo4 (3)**
>
> 8.  **The authors only show the results on heavy-weight models such as ResNet. It would be better for the authors to conduct more experiments on light-weight models such as MobileNet-V2 .**
> We thank the reviewer for the suggestion, and will present results on MobileNet-V2 trained on the CIFAR10 dataset shortly.
>
> 9.  **More ablations of the  η  are required because the influence of η  is unclear. How does this hyper-parameter affect the performance of the proposed pruning method?**
>  $\eta$ is a threshold for determining which filters are discriminative. We define  $\eta$-TV Separability in Definition 4, on Page 5, and use this definition in Algorithms 1 and 2. As we discuss in Section 6.1, the influence of $\eta$ is that it thresholds which filters can be considered discriminative or not, and thus, for a given layer, determines how many discriminative filters can be found in that layer. This therefore determines the extent to which a layer has discriminative filters or not, and consequently determines the extent to which that layer is pruned.
>  Values of $\eta$ that yield effective pruning algorithms vary between datasets and architectures, and even from layer to layer. To address this problem, we introduced IterTVSPrune, or Algorithm 2, which iteratively reduces the threshold $\eta$, while checking against the accuracy on the (partitioned) test set.
> 10.  **The authors only present the experimental results on image classification task. It would be better for the authors to show the experimental results on more computer vision tasks such as object detection and semantic segmentation.**
> We thank the reviewer for this suggestion. However, numerous recent works such as [3,4], produce baselines only for classification tasks. We believe that additional tasks are beyond the scope of our present work, but we endeavour to introduce experiments on alternate machine learning tasks as part of our future work.
>
> We hope that we have addressed the reviewer's concerns, and look forward to future engagement about our work.
>
> **References**
>
> [1] Hoefler et al. *Sparsity in Deep Learning: Pruning and growth for efficient inference and training in neural networks.* 2021.
>
> [2] Blalock et al. *What is the State of Neural Network Pruning?* 2020.
>
> [3] Tanaka et al. *Pruning neural networks without any data by iteratively conserving synaptic flow.* 2020
>
> [4] Sui et al. *CHIP: CHannel Independence-based Pruning for Compact Neural Networks.* 2021
>
> [5] Li et al. *Pruning Filters for Efficient ConvNets.* 2016
>
>
> [6] Tsybakov. *Introduction to Nonparametric Estimation*. 2003.
>
> [7] Shen et al. *Structural Pruning via Latency-Saliency Knapsack*. 2022.

---

> > ### Comment · Reviewer_gKo4 · 2022-12-02
> > **Thanks for the response**
> >
> > Dear authors,
> >
> > Thanks for your detailed responses. The rebuttal has addressed most of my concerns and I would like to keep my score as "weak accept".
> >
> > Best, Reviewer gKo4

---

### Official Review · Reviewer_fFbD · 2022-11-03

**Confidence:** 3
**Correctness:** 4
**Technical Novelty And Significance:** 3
**Empirical Novelty And Significance:** 3
**Recommendation:** 8

**Clarity, Quality, Novelty And Reproducibility:**

The paper is very clear. It explicitly calls out the hypothesis under which the proposed method is designed and provides reasonable intermediary validation steps.

The novelty of the paper lies in designing a cheap pruning method which is cheap due to well validated/referenced assumptions such as Gaussian nature of intermediate activations or strong correlation between discriminability of a layer's output and its prunability.

Authors made appreciable effort to improve reproducibility of the results in the paper.
They provide a link to their implementation.
The authors describe the algorithm in great detail.
They utilize standard architectures to demonstrate their pruning method.
They utilize open datasets for experimental validation.

**Strength And Weaknesses:**

Strengths
------------
The paper clearly calls out the hypothesis and constantly justifies the algorithmic decisions in the context of this hypothesis.
It performs intermediate validation exercises for their hypothesis, which motivate the reader and guide them through the author's intuitions.
The supplementary material is very exhaustive and helpful in further clarifying the details of the proposed algorithm/metrics.

Weaknesses
---------------
The motivation for data free pruning is not clearly described in the paper. The readers are forced to rely on the references.
It would be very informative to the reader to compare the sparsification potential of this technique to a pruning method which exploits fine-tuning post compression.


**Summary Of The Paper:**

In this paper authors propose a mechanism to prune a convolutional neural network model in a relatively data-free manner i.e., they do not utilize training data or loss function for retraining the pruned model. However unlike the actual data-free pruning techniques they assume the availability of moments of class-conditional distributions of the activations. They make a critical assumption that these distributions are Gaussian to exploit cheaply computable sufficient statistics in deriving tractable bounds which are used to guide the pruning process.

They propose a pruning method which exploits the proposed metric to both decide the extent of pruning for a given layer and the actual filters to prune in a given layer, without measuring the impact on the down-stream layer outputs either in terms of the deviations in the metric or in terms of actual performance. This is a reasonable operation under the fundamental hypothesis informing this paper, i.e., discriminability of a filter strongly correlates with performance impact on pruning it. However they do design an iterative version of their pruning method which explicitly measures the cumulative impact of pruning at all the layers on the over all model.

They show appreciable reductions in performance degradations for a given pruning budget compared to other data-free pruning methods.

**Summary Of The Review:**

This paper guides the reader through the design of the algorithm and motivates the design decisions with well validated/referenced hypotheses. It addresses a problem of critical interest to this community. The writing style is clear. The references are more than adequate. The proposed technique is has limited but sufficient experimental validation. Though there are minor possible improvements the paper in its current form is already useful to the readers.

---

### Author Response · Authors · 2022-11-16
**Responses and Updated Manuscript submitted**


We thank the reviewers for their insightful feedback and comments. We have made appropriate changes to the manuscript, either as suggested by reviewers, or inspired by the feedback we received. We have also conducted a variety of additional experiments to address some of the concerns raised by the reviewers. We provide a brief list of manuscript changes and additional experiments below.
1. As requested by the reviewers, we have conducted experiments to evaluate our proposed method in the regime wherein the training set/loss function are available to us, and we are able to retrain the model after pruning. We present a report on these experiments in the new Section E.5 of the supplementary material.
2. We have corrected the typos noticed by Reviewers Ghni and gKo4. Furthermore, we have amended the citation formatting as pointed out by reviewer gKo4. Furthermore, we hope we have clarified the additional concerns regarding definitions, notations, and hyperparameters raised by the reviewers.
4. As requested by reviewer Ghni, we conducted a number of additional experiments to validate the principle hypothesis of the paper, and the scoring mechanism for pruning derived from it. These experiments and results are detailed in section E.6 of the supplementary material.

We hope that our responses have addressed the reviewer's concerns, and look forward to further discussions.

---

### Decision · Program_Chairs · 2023-01-20

**Decision:**

Accept: poster

**Justification For Why Not Higher Score:**

Two reviewers agree to accept the paper, one reviewer weakly accept and one reviewer keeps rejection for some concerns on the claims. The manuscript is thus accepted if the authors made all the promised revision in the final version.

**Justification For Why Not Lower Score:**

Three reviewers agree to accept the paper while one reviewer keeps the negative attitude. All the reviewers agree that the idea in this paper is interesting. Based on the majority reviews, acceptance is suggested.

**Metareview: Summary, Strengths And Weaknesses:**

The manuscript proposes a distributional pruning method to structurally prune deep neural networks. It uses the following hypothesis: Non-discriminative filters do not contribute a lot to the predictive performance of a network and can be removed safely. Specifically, it measures the discriminating ability of filters through the total variation (TV) distance between the class-conditional distributions of the feature maps outputs by the filters. This can be done when a sample from the original distribution is available while the original training data and loss are unavailable. Then, based on the above TV-distance, it defines the LDIFF score to decide the pruning ratio of each layer. These two ideas can be applied iteratively to prune the model for greater sparsity. Experimental results on CIFAR10/100 and ImageNet demonstrate the effectiveness of the proposed method. Such a distributional pruning idea is novel. There are various questions raised by reviewers about the writing and implementation details, where the authors provide clarification and improvement. Three reviewers agree to accept the paper while one reviewer keeps the negative attitude. Based on the majority reviews, the manuscript can be accepted had the authors included all the promised revision in the final version.

**Note From Pc:**

if the above contains the word "oral" or "spotlight" please see: "oral" presentation means -> notable-top-5% and "spotlight" means -> notable-top-25%. As stated in our emails, we are disassociating presentation type from AC recommendations